# Self-replication of DNA by its encoded proteins in liposome-based synthetic cells

Pauline van Nies[1], Ilja Westerlaken[1], Duco Blanken[1], Margarita Salas[2], Mario Mencía[2] & Christophe Danelon[1]

Replication of DNA-encoded information and its conversion into functional proteins are universal properties of life. In an effort toward the construction of a synthetic minimal cell, we implement here the DNA replication machinery of the Φ29 virus in a cell-free gene expression system. Amplification of a linear DNA template by self-encoded, de novo synthesized Φ29 proteins is demonstrated. Complete information transfer is confirmed as the copied DNA can serve as a functional template for gene expression, which can be seen as an autocatalytic DNA replication cycle. These results show how the central dogma of molecular biology can be reconstituted and form a cycle in vitro. Finally, coupled DNA replication and gene expression is compartmentalized inside phospholipid vesicles providing the chassis for evolving functions in a prospective synthetic cell relying on the extant biology.

[1] Department of Bionanoscience, Kavli Institute of Nanoscience, Delft University of Technology, van der Maasweg 9, Delft 2629 HZ, The Netherlands. [2] Centro de Biología Molecular "Severo Ochoa" (CSIC-UAM), Universidad Autónoma, Canto Blanco, Madrid 28049, Spain. Correspondence and requests for materials should be addressed to C.D. (email: c.j.a.danelon@tudelft.nl)

The in vitro construction of a minimal cell from separate biochemical parts (existing or novel) forms one of the outstanding challenges in synthetic biology[1–9]. Unlike abiogenic approaches to develop protocells that solely rely on compounds that were plausibly present on the primordial Earth[7, 10–12], the biosynthetic framework exploits the compositional diversity of the modern biology to assemble a minimal—yet living—cell. Such a minimal cell does not per se have to be a minimized version of an existing organism. Its construction scheme entails the integration of proteins, genes and working mechanisms that are inspired or directly derived from different organisms across the three domains of life, as well as viruses.

As a universal attribute, a living cell—even in its simplest representation—must be able to replicate information to enable proliferation. This information is coded in the form of nucleic acid sequences and must be converted into proteins to support cellular functions. The central dogma of molecular biology formulates the general rules for information transfer[13, 14] and is ubiquitous among all organisms: The genomic DNA is replicated and expressed into non-coding or messenger RNAs (transcription), the latter serving as a template to produce one or more proteins (translation). Hence, meeting the challenge to reconstruct a minimal cell involves the in vitro implementation of DNA replication, transcription and translation. Moreover, compartmentalization is an essential design strategy for coupling genotype and phenotype, while containing the spread of replication parasites. Phospholipid vesicles, called liposomes, with cell-like dimensions may provide such an evolutionary unit.

The development of cell-free protein synthesis (or in vitro transcription-translation, IVTT) systems has provided synthetic biologists with a versatile platform that recapitulates the flow of genetic information from DNA to protein in vitro[15–18]. Of particular relevance is the so-called PURE (Protein synthesis Using Recombinant Elements) system, a well-defined E. coli-based reconstituted protein synthesis platform[19]. Starting from DNA templates, several proteins, including (trans-)membrane proteins, involved in various biological functions have already been synthesized in an active state in the PURE system[20–25].

In parallel, several isothermal DNA replication machineries of increasing level of complexity have successfully been reconstituted from a minimal set of purified proteins. These include the bacteriophage Φ29[26], T7[27], and T4[28] replication complexes, bacterial[29–31] and the yeast[32] replisomes. In the context of building a minimal synthetic cell, the relative simplicity of viral mechanisms compared with their bacterial or eukarial analogues make them attractive candidates. Despite recent progress to couple DNA replication and gene expression[33, 34], an integrative system, whereby the parental double-stranded DNA template is faithfully replicated by self-encoded proteins, has not been realized yet. This is a major obstacle toward a self-reproducing evolvable synthetic cell.

Here we approach this challenge by first identifying the Φ29 DNA replication mechanism as a promising candidate. The Φ29 DNA polymerase has already been suggested for replication of a minimal genome via a process combining rolling circle amplification of a circular genome and subsequent circularization of the linear DNA product through recombination[2]. However, this mechanism has so far failed to work[34] and the replication of DNA by DNA-encoded proteins—a hallmark of cellular life—has remained elusive. In contrast to earlier strategies, we propose herein to use the protein-primed replication mechanism of the Φ29 virus to amplify a linear synthetic genome. Protein-primed DNA synthesis is an elegant solution to the initiation/termination paradox as replication is not accompanied by loss-of-sequence information[35]. Moreover, the DNA product of replication is a copy of the template, hence no further DNA processing is necessary to regenerate the parental DNA. The Φ29 genome is a linear 19.3-kb double-stranded DNA that is transcribed by the host Bacillus subtilis RNA polymerase to produce its own replication proteins. Amplification of the Φ29 genome occurs in a symmetrical manner, the initiation of replication taking place at the left and right terminal regions by a protein-primed mechanism[35]. In vitro, functional origins of replication consist of 194-bp or 68-bp (minimal origin) sequences and a phosphate group is required at the 5′ end. The minimal set of Φ29 replication proteins required for in vitro amplification of heterologous DNAs consists of the terminal protein (TP, p3), DNA polymerase (DNAP, p2), single-stranded DNA binding protein (SSB, p5) and double-stranded DNA binding protein (DSB, p6)[36].

The TP and DNAP form a 1:1 complex in solution, whose stable assembly is stimulated by the ammonium ion[37]. The Φ29 replication origins include a specific end sequence that is recognized by the DNAP-TP complex and high-affinity binding sequences for the DSB p6. The hydroxyl group of a specific serine in the terminal protein p3 serves as an acceptor of the first dNMP creating a covalent linkage with the 5′ end of the DNA. The Φ29 DNA polymerase p2 is a highly processive polymerase with strand displacement and proofreading activities, ensuring continuous elongation from the 5′-end to the 3′-end of the linear DNA. Recruitment of the p2–p3 heterodimer to a Φ29 replication origin is enhanced by the presence of parental TP. Protein p5 is known to coat the displaced ssDNA preventing its degradation by host nucleases, as well as avoiding strand switching by the DNAP in vitro[38]. Protein p6 binds with high affinity to the Φ29 DNA replication origins, distorting the DNA helix and stimulating the incorporation of the first nucleotides[39].

Our objective was to demonstrate that the minimal set of Φ29 replication proteins can be synthesized from their genes in the PURE system and can collectively amplify their own DNA template without loss- of information, completing one round of the central dogma in biology. Eventually, one-pot DNA, mRNA and protein synthesis was implemented inside liposome-based synthetic cell models.

## Results

**Expression of linear DNAs ended with Φ29 replication origins**. Our design strategy for the DNA templates relies on a number of requirements regarding both the Φ29-based replication mechanism and the regulatory elements for gene expression in the PURE system (PUREfrex) (Fig. 1a–c, Supplementary Note 1, Supplementary Fig. 1). To implement a coupled in vitro transcription-translation-DNA replication (IVTTR) cycle, the DNA template coding for the Φ29 replication proteins was engineered with the 194-bp terminal origins of replication of Φ29 (Supplementary Table 4). Here, the DNA serves as a template for both the transcription and replication processes (Fig. 1a). Alternatively, DNA templates that do not contain the origin ends support production of the Φ29 replication proteins that can be directed to a different target amplification DNA like the Φ29 genome. We verified that the full-length proteins can be successfully synthesized from their DNA templates in the PURE system (Fig. 1d, e). Furthermore, co-expression of all four replication proteins starting from two dual-gene DNA templates was confirmed (Fig. 1e).

**The in vitro synthesized Φ29 replication proteins are active**. First, we confirmed that the synthesized DNAP retains its functional properties in PUREfrex reactions, including its strand displacement activity (Supplementary Methods, Supplementary Note 3 and Supplementary Fig. 4-6). Next, we investigated under which conditions the co-synthesized proteins are able to

collectively support efficient amplification of the Φ29 genome in one-pot IVTTR reactions. The conditions for in vitro replication of the Φ29 genome by the purified Φ29 DNA synthesis machinery have been well established[36]. Challenging the de novo synthesized replication machinery to amplify the ~19-kb Φ29 genome is important in the prospect of replicating a complete minimal genome (see Discussion). The Φ29 replication proteins were encoded on linear DNA templates that did not contain the replication origins, so that the machinery is exclusively directed to the Φ29 genome. Alternatively, DNA constructs flanked with origins can be used at subnanomolar concentration to bias replication toward the Φ29 genome. Co-expression of DNAP and TP led to eightfold amplification of the Φ29 genome after 4 h (Fig. 2, Supplementary Note 5), similar to the yield obtained with

40 ng of each purified protein, corresponding to 30 nM of DNAP and 65 nM of TP (Supplementary Fig. 7). Under these conditions, some immature elongation products are generated, visible as a smear or lower bands on gel. This is also observed with purified proteins (Supplementary Fig. 7). Additional expression of the *p5* gene does not result in a higher production of DNA, but it reduces the formation of side products, thus improving the overall quality of the newly synthesized DNA (Fig. 2). This result suggests that the amount of synthesized p5 protein is high enough to assist the DNAP-catalyzed polymerization and enhance faithful amplification of dsDNA by coating the strand-displaced ssDNA. Alternatively, the positive effect of *p5* co-expression could be attributed to a lower concentration of the synthesized DNAP and TP, which results from the sharing of the

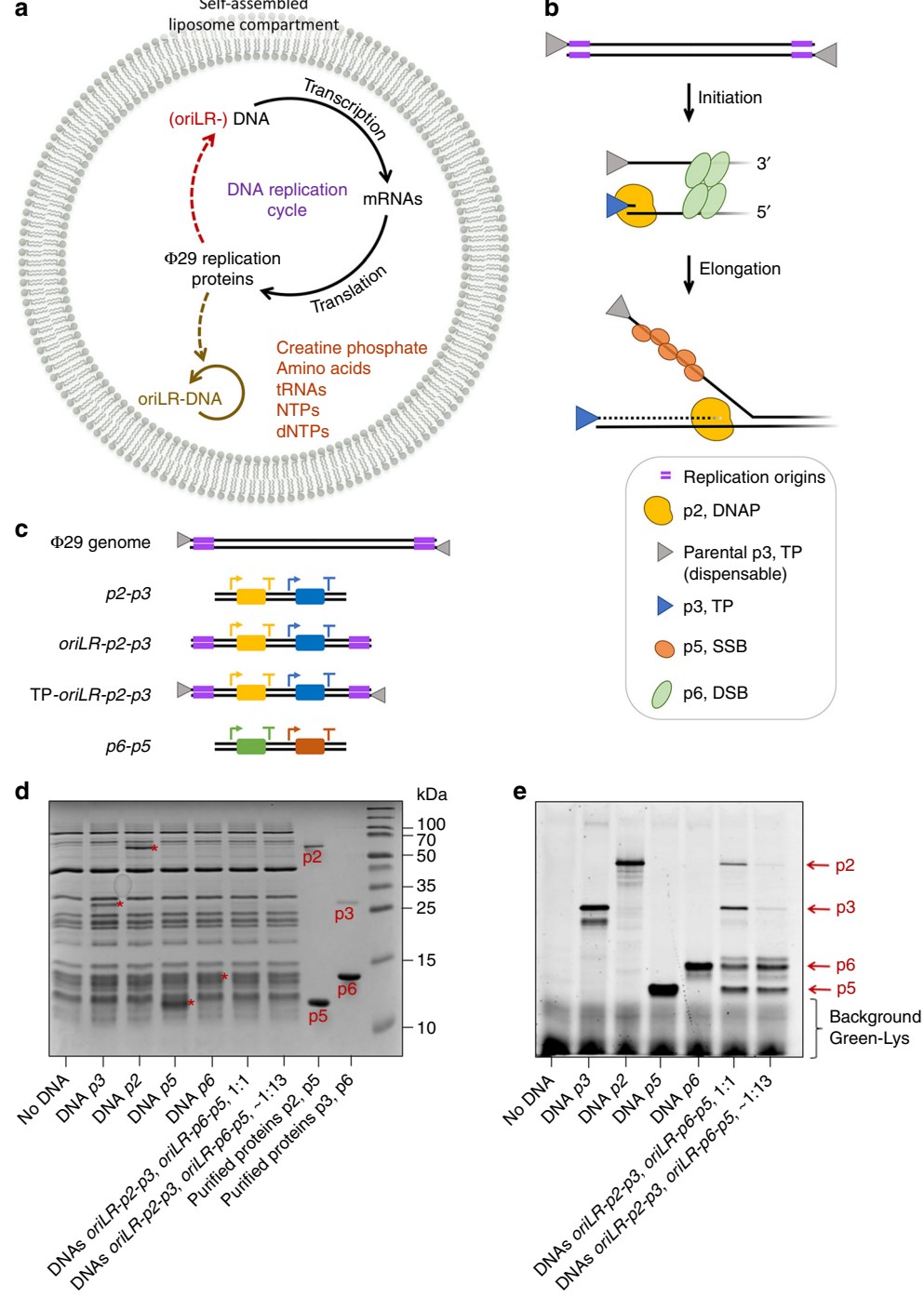

transcription/translation resources and machinery with an extra gene. Hence, the concentration of DNAP-TP complex should not exceed the amount required for optimal amplification[38]. Co-expression of the *p6* template only or of the *oriLR-p6-p5* dual-gene construct yields ~6-fold and ~4-fold amplification of the Φ29 genome, respectively, with reduced formation of short-sized replication products. Whereas the minimal set of proteins for DNA replication comprises the DNAP and TP, the presence of the p5 or/and p6 auxiliary proteins diminishes the formation of side products. Together, these results show that a gene-encoded, fully synthetic replication machinery, is capable to support replication of the TP-bound Φ29 genome.

**Gene encoded proteins that amplify their own DNA template**. Replication of gene-encoding DNA without loss of information is a prerequisite for faithful copying of a minimal genome and maintenance of functions. To implement a full DNA replication cycle (Fig. 1a), we aimed to amplify the *oriLR-p2-p3* DNA coding for the minimal Φ29 replication system, namely the DNAP and TP proteins (Fig. 3a). The *oriLR-p2-p3* template containing the Φ29 sequence ends (or without in control reactions) was tested for expression in the PURE system. Whereas modest (≤2-fold) amplification of the full-length DNA substrate was measured with the synthesized p2 and p3 only, addition of purified p5 and p6 greatly improved replication efficiency up to ~20 times, while reducing the amount of short products (Fig. 3b). Optimizing the gene expression conditions further potentiated replication and a ≥50-fold amplification factor could be reached (Supplementary Fig. 9 and Supplementary Note 5). Co-expressing the *p5* and *p6* genes failed to yield high amplification (Supplementary Fig. 9, 10), presumably because the concentrations of p5 and p6 were too low in the first critical minutes of the reaction under this four-gene IVTT condition (Fig. 1e, Supplementary Fig. 3). As expected, in the absence of the Φ29 origins (*p2-p3* construct), no replication of the full-length DNA was clearly visible, even in the presence of purified p5 and p6 (Fig. 3b). However, occurrence of replication initiation events is manifested by the appearance of lower DNA bands on gel. These results imply that the *p2-p3* template can be used to direct the synthesized replication machinery to a different target DNA without severely competing for resources.

To confirm completion of a full round of the central dogma of biology, the newly synthesized DNA was purified and its ability to serve as a template for transcription and subsequently generate the encoded full-length proteins, was assayed. First, the end-point IVTTR reaction solution was treated with λ-exonuclease to eliminate the DNA that is not linked to TP (only the TP-capped

DNA is resistant to nuclease digestion). Full-length and lower molecular mass DNA replication products survived the treatment, whereas the input DNA was successfully eliminated (Fig. 3c). As a last step, the remaining (resistant) DNA was used as input template for a new IVTT reaction and the translation products were visualized on gel (Fig. 3d). Larger amounts of both p2 and p3 proteins were generated when starting from a replication reaction for the *oriLR-p2-p3* DNA assisted by purified p5/p6 proteins. However, the low quantity of amplified full-length DNA when either purified p5/p6 proteins or the *ori* sequences were omitted, was sufficient to unambiguously produce p2 and p3 proteins. These results demonstrate that the DNA-encoded information is amplified and propagates at the transcription and translation levels to produce the expected phenotype of synthesized proteins.

To circumvent the need of using purified p5 and p6 without compromising the amplification yield too much, we prepared purified TP-bound *oriLR-p2-p3* DNA, named TP-*oriLR-p2-p3* (Fig. 4a). Pre-bound TP is dispensable for in vitro amplification of heterologous DNAs[36]; however, it is known to enhance recruitment of the p2-p3 heterodimer initiation complex[36]. In analogy, we reasoned that the parental TP protein on the TP-*oriLR-p2-p3* template will help recruit the de novo synthesized TP and DNAP, and potentiate replication. As expected, introduction of parental TPs remarkably stimulates DNA replication activity in a second IVTTR reaction (Fig. 4b). After 4 h, the full-length DNA product represents a ~20-fold amplification of the input DNA, compared to ≤2-fold increase without pre-bound TP (Fig. 3b). A large amount of short reaction products is also produced (Fig. 4b), denoting an ineffective usage of the replication resources. Addition of 15 µM of purified p5 did not increase the fraction of full-length product. Furthermore, activity of a fully de novo synthesized replication system was demonstrated by co-expressing the TP-*oriLR-p2-p3* template and the *p6-p5* DNA. A ~10-fold amplification of the replication template was measured, while the fraction of short products is reduced to less than ~50% of the total DNA synthesized (Fig. 4b) compared to a reaction merely comprising the p2 and p3 proteins. We interpret the reduction of side products as a manifestation of the functionality of the synthesized p5 and p6 proteins. Because these short-sized replicons survive λ-exonuclease treatment, they must be TP-bound. Importantly, the TP-*oriLR-p2-p3* used here as a template was itself produced from the encoded p2 and p3 proteins in a previous IVTTR reaction (Supplementary Fig. 12). Therefore, amplification of the TP-*oriLR-p2-p3* DNA, as shown in Fig. 4b, demonstrates that the copied DNA encodes for active proteins. To push further the autocatalytic DNA replication cycle (Fig. 4e), the ability of the newly synthesized DNA to support expression of

**Fig. 1** Basic elements of the DNA self-replication strategy. **a** Flow of genetic information reconstituted in batch mode reaction or inside a liposome. The Φ29 virus-inspired DNA replication mechanism is implemented in the PURE*frex* transcription-translation system. A DNA replication cycle is completed when the DNA template expressing the Φ29 proteins is also the replicating DNA. Alternatively, the expressing DNA does not contain the Φ29 origin sequences (oriLR- in brackets) and a different target DNA is used as a replication substrate (solid colored line). Some essential reaction substrates are indicated. **b** Schematic depicting the mechanism of replication initiation by the Φ29 system. **c** Schematic of the Φ29 genome and four de novo designed DNA constructs used in this study (Supplementary Table 1). The most relevant regulatory elements are depicted: the T7 promoter (arrows), the vesicular stomatitis virus (VSV) internal terminator[47] or the T7 terminator at the 3′ end (T, for both terminators), the genes (rectangles) and the Φ29 origins of replication. Their termination efficiency was experimentally estimated (Supplementary Fig. 2). **d**, **e** Analysis of translation products on polyacrylamide protein gels. PURE*frex* solution was supplemented with BODIPY-Lys-tRNA$_{Lys}$ (GreenLys) to introduce fluorescent lysine residues in the synthesized proteins and distinguish them from the IVTT protein background (**e**). Coomassie Blue staining was also performed to visualize the purified proteins and the total protein content in PURE*frex* reactions (**d**). Amounts of purified proteins: 180 ng p2, 2 µg p5, 180 ng p3, and 2 µg p6. The estimated concentrations are 1.0 µM for DNAP, 4.0 µM for TP, 5.0 µM for p5 and 1.7 µM for p6, when all genes are separately expressed (Supplementary Note 4 and Supplementary Fig. 3). The production of all four full-length proteins was confirmed when the *oriLR-p2-p3* and *oriLR-p6-p5* templates were co-expressed in equimolar amounts (1:1) or with an excess of the *oriLR-p6-p5* DNA (~1:13), the latter ratio being used in replication experiments, where larger amounts of p5 and p6 are required. Note also the generation of truncated translation products, in particular for p2 and p3. Predicted molecular masses are 12 kDa for p6, 13.3 kDa for p5, 31 kDa for p3, and 66 kDa for p2

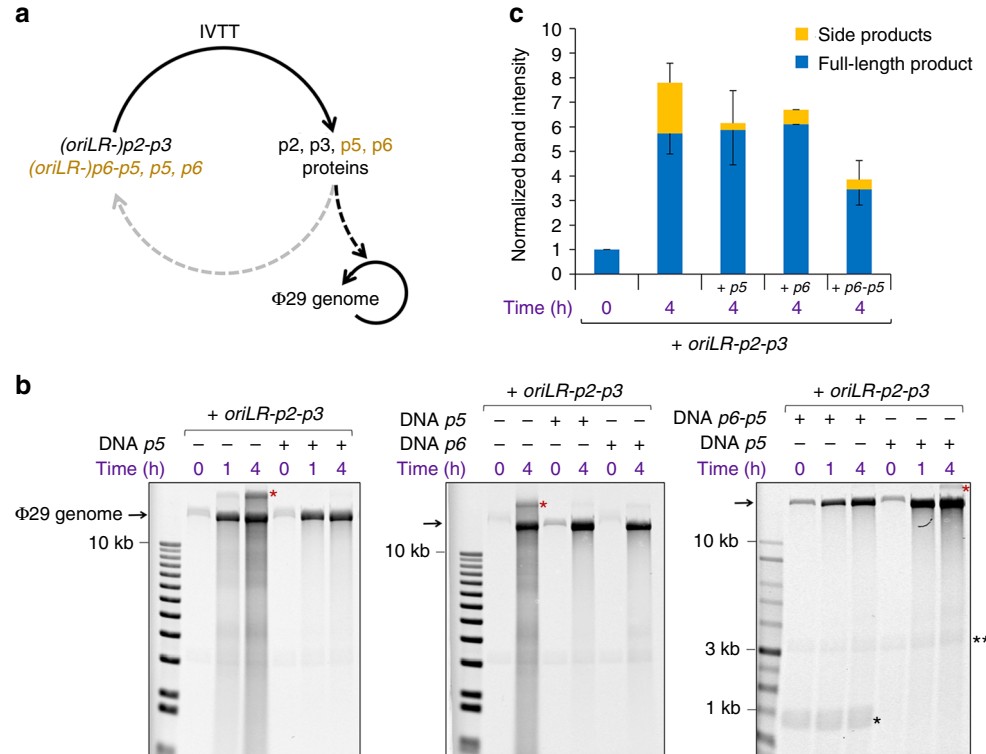

**Fig. 2** Replication of the Φ29 genome with de novo synthesized proteins. **a** Reaction pathways for gene expression (IVTT) and DNA amplification. The replication machinery is preferentially directed to the TP-capped Φ29 genome (black arrows). Co-synthesis of the p5 and p6 proteins from their genes is also indicated. **b** Alkaline agarose gels of the expression-amplification reaction products under various experimental conditions. The p2 and p3 proteins were produced from the *oriLR-p2-p3* DNA. The p5 and p6 proteins were expressed from the *p5*, *p6*, or *oriLR-p6-p5* genes. Under these conditions (about equimolar amounts of input Φ29 genome and lower-mass *oriLR-p2-p3* DNA), replication is strongly biased toward the natural TP-bound Φ29 genome. The input Φ29 genome can be seen at time zero, while the *oriLR-p2-p3* and *oriLR-p6-p5* DNAs are visible in some gels (indicated as double and single black asterisks, respectively). The red asterisk indicates the upper band of the Φ29 genome, as also observed with the stock DNA (Supplementary Fig. 8) and after amplification by the purified proteins (Supplementary Fig. 7). First lane on gels is the DNA ladder. **c** Quantitative analysis of the experiments shown in **b**. Values represent the mean and standard deviation (sdv) from three independent experiments. For clarity, only the negative or positive sdv error bars are represented for the full-length product and side products, respectively

the encoded proteins was confirmed by performing a third IVTT reaction using the second-generation amplification products as input templates. Clear bands representing full-length synthesized p2 and p3 can be observed (Fig. 4d). These results demonstrate that DNA replication is potentiated by pre-bound TP and that this process maintains the encoded gene function despite the production of short replicons.

**Coupled DNA replication and IVTT in liposomes.** Evolution of DNA-encoded proteins requires a linkage between the genetic and phenotypic components. We used phospholipid vesicles (liposomes) as the compartment of the IVTTR-based synthetic cell (Fig. 1a). The liposome bilayer is composed of a mixture of biologically relevant lipids that mimic the inner surface of the *E. coli* cellular membrane. We first validated that the complete chain of reactions to convert a DNA program into active proteins can be reconstituted in cell-sized compartments. Direct visualization of translation activity was ensured by synthesizing from its gene the yellow fluorescent protein (YFP). Thousands of micrometer-sized liposomes (here labeled with a red membrane dye) encapsulating the PURE system were produced (Fig. 5a). A large fraction of vesicles is unilamellar with a diameter ranging from ≤1 μm up to 15 μm. Tens, up to hundreds, of liposomes successfully express YFP in a single field of view, representing about 30% of the total vesicles. We then demonstrated that the in

vesiculo synthesized p2 can elongate a primer-template junction and produce a transcriptionally active dsDNA template (Supplementary Methods and Supplementary Fig. 17).

Next, we examined whether expression of the *oriLR-p2-p3* gene could provide the basis for a full IVTTR cycle inside liposomes. To increase the yield of DNA replication, the reaction was supplemented with purified p5 and p6 proteins. Acridine orange was chosen as a DNA intercalating dye (Supplementary Note 6). Figure 5b shows that expression of the *oriLR-p2-p3* template yields an increase of the fluorescence intensity inside the liposome lumen. Interestingly, the signal is not evenly distributed but appears as a bright fluorescent spot. Control experiments where dNTPs were omitted fail to produce such DNA fluorescence signal (Fig. 5b), indicating that DNA replication causes accumulation of acridine orange, and not the RNA and input DNA background. These results demonstrate that the central dogma of molecular biology, whereby DNA replication, transcription and translation are coupled, can be reconstituted in closed liposomal evolutionary units.

Encouraged by the high-amplification yield obtained by co-expressing the TP-capped TP-*oriLR-p2-p3* replication template and *p5-p6* genes in bulk reactions (Fig. 4b), we sought to reproduce this experiment in liposome-confined reactions. No notable increase of the acridine orange fluorescence intensity over the background was observed when the reaction was supplied with dNTPs (Supplementary Fig. 14). This result suggests that

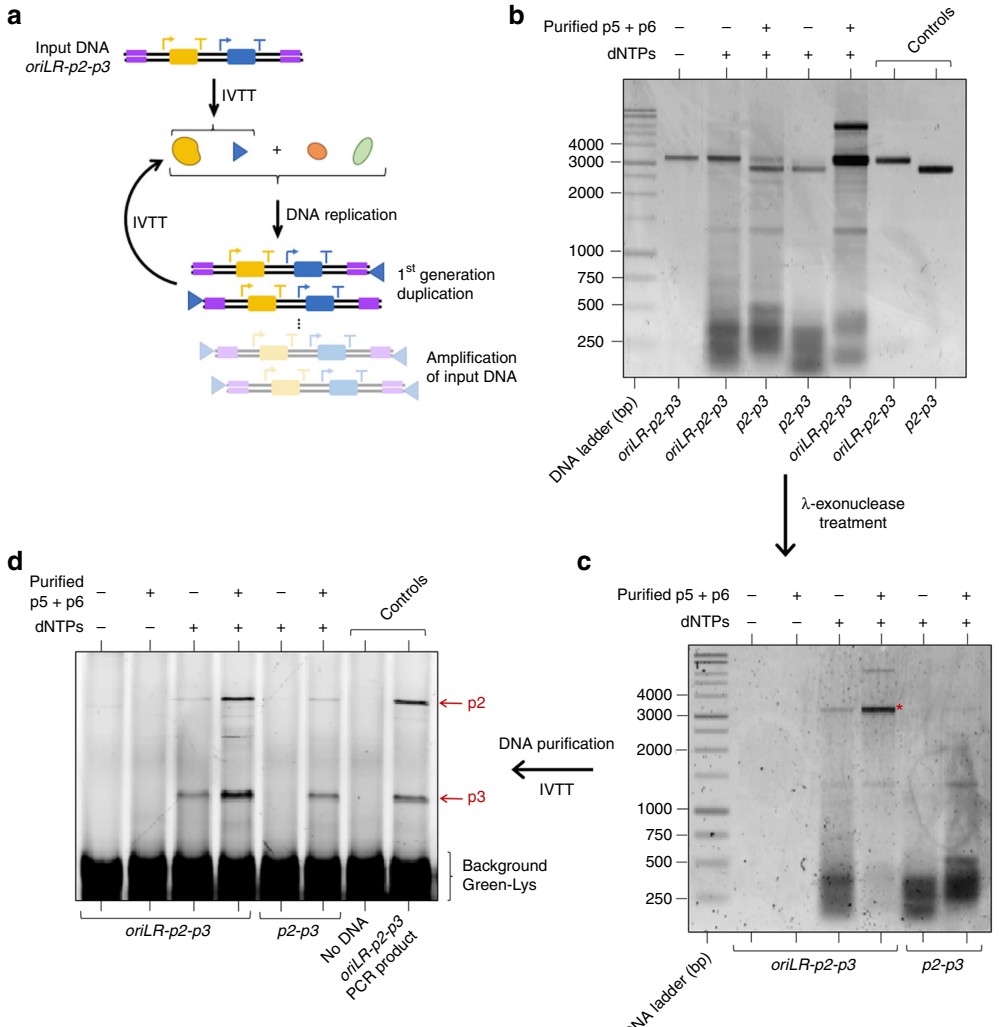

**Fig. 3** Replication of DNA by its encoded proteins. **a** IVTTR reaction scheme using the *oriLR-p2-p3* DNA template. Short amplification products are not represented. **b** The replication products of either the *oriLR-p2-p3* or the *p2-p3* DNA template (100 ng input) expressed in PURE*frex* were visualized on agarose gel after RNase and Proteinase K treatments, followed by RNeasy clean-up column purification. The results from five independent replication experiments are shown in Supplementary Fig. 9a, Supplementary Fig. 10 and Supplementary Fig. 12b,e. In each IVTTR reaction triggered by the expression of the *oriLR-p2-p3* DNA construct, 2.5 nM of template produced about 100 nM of p2 and 700 nM of p3 proteins (as estimated in Supplementary Fig. 3), which were able to generate ~50 nM of full-length DNA product when the reaction was supplemented with purified p5 and p6. **c** Samples were further incubated with λ-exonuclease to remove TP-uncapped DNA. The asterisk indicates full-length TP-capped DNA that has not been degraded by the λ-exonuclease. **d** De novo synthesized DNA was subsequently used as a template for a second IVTT reaction. The translation products were visualized by PAGE with GreenLys labeling. Expression of DNA that resulted from an IVTTR in the presence of purified p5 and p6 proteins led to fluorescent p2 and p3 protein bands of similar intensity as that measured when starting with 2.5 nM purified DNA (control with PCR product) demonstrating that the encoded functions are retained during amplification. Protein gels from two independent replication experiments are shown in Supplementary Fig. 9b and Supplementary Fig. 11. Note that the modest replication efficiency in the absence of purified p5 and p6 was sufficient to generate the encoded p2 and p3 proteins through amplification of information at the transcription and, to a lower extent, at the translation levels

fine-tuning of the four-gene expression conditions in liposomes and perhaps optimization of DNA labeling will be necessary to obtain visible replication of the TP-*oriLR-p2-p3* substrate.

## Discussion

We demonstrated here that the Φ29 DNA replication machinery is a promising approach to replicate genomic DNA in model synthetic cells relying on the modern biology (see Supplementary Note 7 for a discussion about alternative mechanisms). The basic requirements of faithful replication of a protein-encoding DNA template and its encapsulation in cell-sized liposomes have successfully been fulfilled. This recapitulates the three central

processes of the flow of genetic information, whereby the newly replicated dsDNA feeds back new IVTTR cycles.

We here discuss current limitations and prospects for further improvements of the Φ29 DNA replication machinery coupled to gene-encoded synthesis with the PURE system. As previously reported, the presence of ribonucleotides (NTPs) and tRNAs can have inhibitory effects on polymerization by the Φ29 DNAP[34, 40]. We also found that the presence of NTPs at concentrations similar to that in PURE*frex* significantly slows down DNAP activity (Supplementary Fig. 15, 16). Though amplification of the parental DNA genome into a large copy number is likely unnecessary (even detrimental) for vertical inheritance in proliferating synthetic cells, fine-tuning the ratio of ribo- and

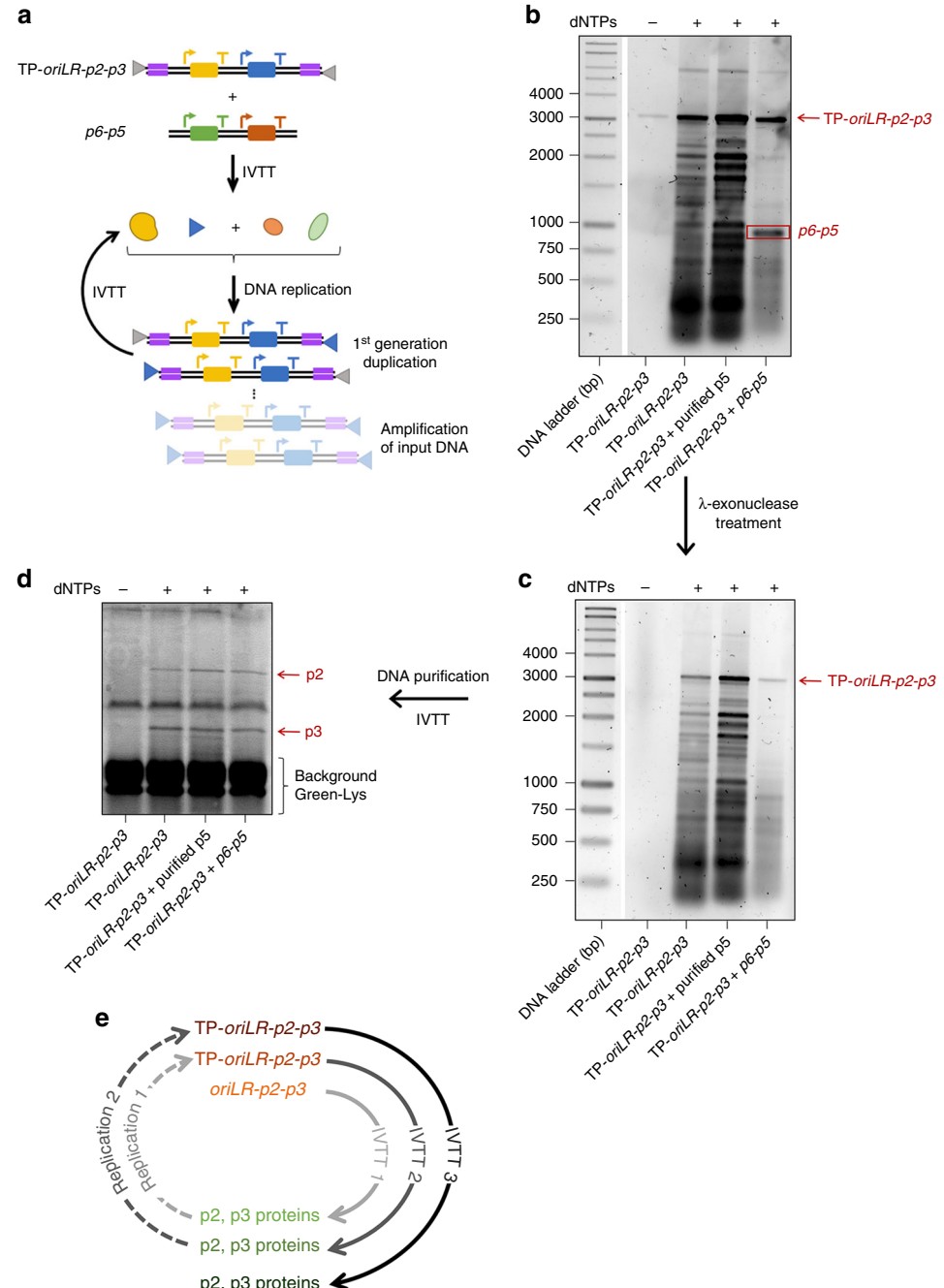

**Fig. 4** Potentiating DNA self-replication with 5′-end pre-bound TP. **a** IVTTR reaction scheme using the TP-*oriLR-p2-p3* DNA template. Short amplification products are not represented. The detailed experimental workflow, including preparation of the TP-*oriLR-p2-p3* DNA, is shown in Supplementary Fig. 12a. **b** The replication products of the TP-*oriLR-p2-p3* DNA template (∼75 ng input, equiv. ∼1.9 nM) expressed in PURE*frex* were visualized on agarose gel after RNase and Proteinase K treatments, followed by RNeasy clean-up column purification. When indicated the *p6-p5* DNA (70 ng input, equiv. ∼5.7 nM) was co-expressed. The results from two independent IVTTR experiments are shown in Supplementary Fig. 12c, f. For direct comparison of the amplification yield with and without parental TP, similar amounts of input DNA were used, the end-point reaction solutions were loaded on the same gel and the band intensities were analysed (Supplementary Fig.13). Clearly, replication of the TP-*oriLR-p2-p3* DNA template is more efficient. **c** Samples were further incubated with λ-exonuclease to remove TP-uncapped DNA. Note that the overall amount of DNA on the gel is reduced (to the extent that the band corresponding to the input TP-*oriLR-p2-p3* DNA in the −dNTPs control sample is no longer visible) after nuclease treatment due to dilution during the cleaning/purification steps. **d** De novo synthesized DNA was subsequently used as a template for a third IVTT reaction. The translation products were visualized by PAGE with GreenLys labeling. The protein gel analysis from an independent IVTTR experiment is shown in Supplementary Fig. 12d. **e** Autocatalytic IVTTR cycles realized in this study. A first IVTTR reaction was performed using *oriLR-p2-p3* as input DNA and producing larger amount of TP-*oriLR-p2-p3* (Supplementary Fig. 12b, e). The purified TP-*oriLR-p2-p3* DNA was subsequently used as template for a second IVTTR (**b**). Finally, the purified DNA products from IVTTR 2 was used for a third IVTT (**d**)

deoxyribonucleotides might be important for sustained synthesis of DNA, mRNA and protein. Concentration of input DNA and temperature are other factors that influence DNA amplification and formation of short replication products (Supplementary Fig. 4d), and they need to be carefully adjusted. Inefficient replication is manifested by the formation of short products. They are likely truncated replicons that accumulate over the time course of the reaction and compete with the full-length template for the replication resources. The symmetrical nature of the Φ29 replication process makes it more susceptible to head-on collision events than asymmetrical mechanisms. Collision might occur between two Φ29 DNAPs polymerizing in opposite directions or

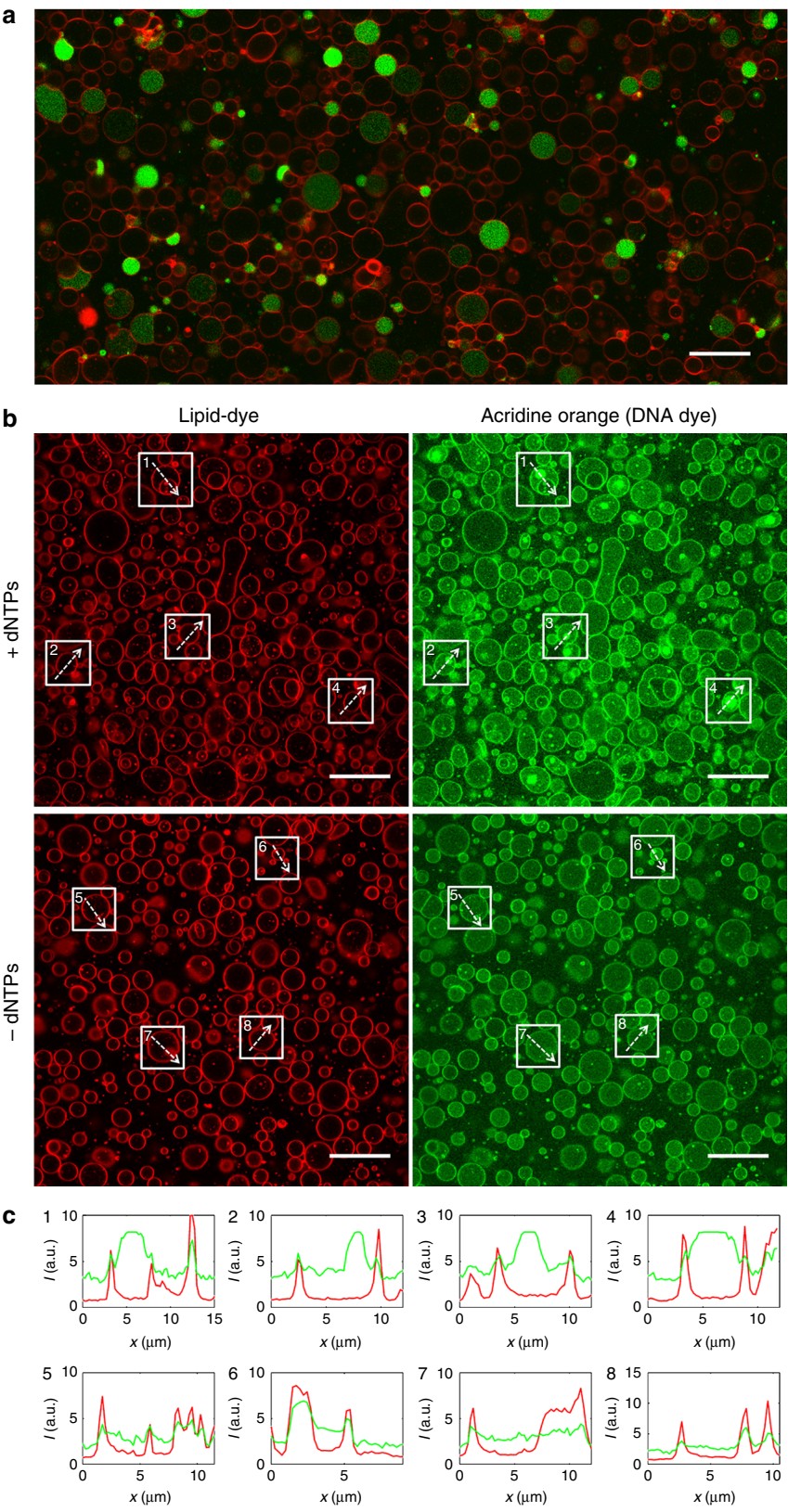

between Φ29 DNAP and other DNA processing enzymes like the T7RNAP[41]. Even though most collision events are expected to have no adverse effects, optimization of the concentrations of the various enzymes and substrates will be key to concomitantly improve efficiency of the different processes. Furthermore, one shortcoming of the PURE system is its poor performance to synthesize large (micromolar) protein concentrations when starting from multiple genes. This is for instance manifested by the fact that co-expression of the *p2*, *p3*, *p5* and *p6* genes failed to produce enough of the p5 and p6 proteins to stimulate replication of a DNA template that is not linked to parental TP (Fig. 3). Possible solutions to bypass this inherent limitation include downregulation of the DNAP and TP protein synthesis, at the benefit of the expression of p5 and p6. This could be achieved at the transcription level by using low- and high-affinity promoters to control expression of the *p2-p3* and *p6-p5* genes, respectively. At the translation level, low- and high-affinity ribosome binding site sequences could similarly be employed. A more radical, and challenging, strategy is to boost the performance of IVTT by optimizing its composition[42] or by implementing continuous-flow reactions[43].

Of note, amplification of the *oriLR-p2-p3* DNA inside liposome results in punctuated accumulation of the acridine orange dye, indicating DNA aggregation. As a possible molecular origin, we propose that spermidine, a compound present in the PURE system and known as a DNA-condensing agent[44], could play a role. Alternatively, accumulation of inorganic phosphate (despite the presence of a pyrophosphatase in the PURE system) during transcription and replication combined with the presence of magnesium ions may trigger the formation of DNA aggregates[45]. Although the precise mechanism requires further elucidation, this observation may be pertinent in the prospect of DNA partitioning during synthetic cell division.

The key combination of genomic DNA replication by encoded proteins and its compartmentalization inside lipid vesicles, as demonstrated here, provides the basic units to evolve functions in a synthetic minimal cell (Fig. 6). One can envisage that random mutations occurring during DNA replication through activity of the wild-type Φ29 DNAP or of an engineered error-prone mutant would lead to diversity generation. New phenotypic traits with a selective advantage might appear, for instance in the form of a faster and more robust DNA replication system. One can also envision how to evolve functions beyond DNA replication, by incorporating the relevant genes for lipid biosynthesis[25] and vesicle division (Fig. 6), enabling improved interfacing between the subsystems originating from different organisms.

A ribosome-based minimal cell, i.e., based on the extant biology but made of a limited number of components, would entail a ~110-kbp genome (corresponding to ~150 genes)[2]. This number might however be underestimated as suggested by a recently engineered minimal bacterium containing 473 genes, of which 149 have unknown function[46]. Given the exceptional processivity of the Φ29 DNAP—over 70 kb[40]—the Φ29-based replication strategy described here is potentially viable to amplify a linear minimal ~150-kbp genome split over two equally sized linear DNA molecules. In its present form, the de novo synthesized machinery can already replicate a ~20-kbp genome (Fig. 2) offering the possibility to introduce ~20 genes. The fact that the protein-primed DNA replication mechanism of phage Φ29 is composed of only four proteins (vs. >12 for the bacterial systems[31, 32]) is an advantage because its expression involves low usage of resources.

## Methods

**Preparation of DNA constructs.** The DNA sequences were submitted to Gen-Script (United States) for gene synthesis and were returned in the pUC57 vector with EcoRV cloning sites. The full sequences for *oriLR-p6* and *oriLR-p6-p5* were directly ordered at the company. The *p2* and *p3* genes were ordered separately and the fusion *p2-p3* constructs with the VSV-repeat terminator[47] were generated by assembly PCR. The list and sequences of the PCR and sequencing primers used in this study are reported in Supplementary Tables 2 and 3. Regular PCR reactions were performed with 1–10 ng of plasmid or linear DNA as template, 1 unit of Phusion polymerase (Finnzymes) in HF buffer containing 0.2 mM dNTPs, 0.2 μM forward and 0.2 μM reverse primers in a final volume of 50 μL. After an initial heating step for 30 s at 98 °C, the PCR reactions consisted of 30 cycles of 10 s melting the DNA at 98 °C, followed by hybridization of the primers for 15 s at 60 °C, and elongation by the DNAP at 72 °C for 30 s per kb template. After the 30 cycles, the temperature remains constant at 72 °C for 5 min to allow the DNAP to complete all remaining polymerization reactions. The PCR-generated linear DNA fragments were purified with the PCR clean-up kit from Promega according to the manufacturer's protocol. The concentration of the purified DNA was measured on the NanoDrop (Thermo Scientific) and the purity of the DNA products was checked on a TAE 0.7–1.1% agarose gel using 100 ng of DNA and ethidium bromide (EtBr) staining. The BenchTop 1-kb DNA Ladder from Promega or the 1-kb Plus DNA ladder from Thermo Fischer Scientific was used to confirm the correct size of the dsDNA templates.

**Purified Φ29 DNA replication proteins.** The purified Φ29 replication proteins consist of p2[48], p3[36], p5[49] and p6[36]. Quantification was done by the Lowry method using Coomasie staining of SDS-PAGE gels with appropriate standards followed by densitometry of the bands. In the case of the DNA polymerase, absolute concentration was also assessed by measuring polymerization activity against known specific activity. Stock concentrations and storage buffers are: p2 (320 ng/μL in 50 mM Tris, pH 7.5, 0.5 M NaCl, 1 mM EDTA, 7 mM 2-mercaptoethanol (BME), 50% glycerol), p3 (400 ng/μL in 25 mM Tris, pH 7.5, 0.5 M NaCl, 1 mM EDTA, 7 mM BME, 0.025% Tween 20, 50% glycerol), p5 (10 mg/mL in 50 mM Tris, pH 7.5, 60 mM ammonium sulfate, 1 mM EDTA, 7 mM BME, 50% glycerol), p6 (10 mg/ml in 50 mM Tris, pH 7.5, 0.1 M ammonium sulfate, 1 mM EDTA, 7 mM BME, 50% glycerol). The proteins were aliquoted and stored at −80 °C. The p2 and p3 proteins were diluted before immediate use into an intermediate solution consisting of 25 mM Tris, pH 7.5, 0.1 M NaCl, and 0.05% Tween 20. The stock solution of the Φ29 genome was 190 ng/μL in 50 mM Tris, pH 7.5, 0.2 M NaCl, 1 mM EDTA, 7 mM BME, 0.05% Tween 20, 50% glycerol.

**Fig. 5** Compartmentalization of self-encoded DNA replication inside liposomes. Images in **a** and **b** display fluorescence confocal micrographs of PURE*frex*-containing phospholipid vesicles labeled with a membrane dye (red). **a** Fluorescence emitted by the YFP synthesized from its gene (7.4 nM bulk concentration of dsDNA) is visualized in green (overlaid channels). Assuming that the entrapped DNA molecules follow a Poisson distribution, liposomes with a diameter of 4 μm contain ~140 DNA copies on average. Thousands of gene-expressing liposomes can be imaged per sample. About 30% of the liposomes produce YFP at a detectable level. This functional heterogeneity is probably a consequence of the compositional diversity of the biochemical network within vesicles. Scale bar is 20 μm. **b** Following the IVTTR reaction scheme shown in Fig. 3a, 5 nM of the *oriLR-p2-p3* DNA template along with the purified p5 and p6 proteins were co-encapsulated with or without dNTPs during liposome formation. After gene expression and liposome immobilization, the DNA staining fluorophore acridine orange (green channel) was injected. Amplification of DNA in the lumen of individual vesicles is accompanied by a higher fluorescence signal of acridine orange in the form of bright spots. We noticed that acridine orange can stain the liposome membrane, presumably due to the hydrophobic nature of its aromatic groups. Nonetheless, the DNA and membrane signals can easily be discriminated by using the red membrane dye for co-localization analysis, so that the lumen signal from amplified DNA can unambiguously be ascribed. The fluorescence images show representative fields of view from three independent experiments. Five different fields of view of similar liposome density were analysed per experiment to quantify the number of 'nucleoids'. Comparing + dNTPs and –dNTPs (+/–) in the three experimental repeats, 350/9, 130/2, and 773/58 nucleoid-like structures were identified. Scale bars represent 20 μm. **c** Line intensity profiles from eight liposomes framed in **b**. In the images from the –dNTPs sample, we deliberately chose liposomes exhibiting green spots to show that they co-localize with the membrane dye, demonstrating that they are of different nature than those triggered by DNA replication. Color coding is the same as in **b**. a.u., arbitrary units

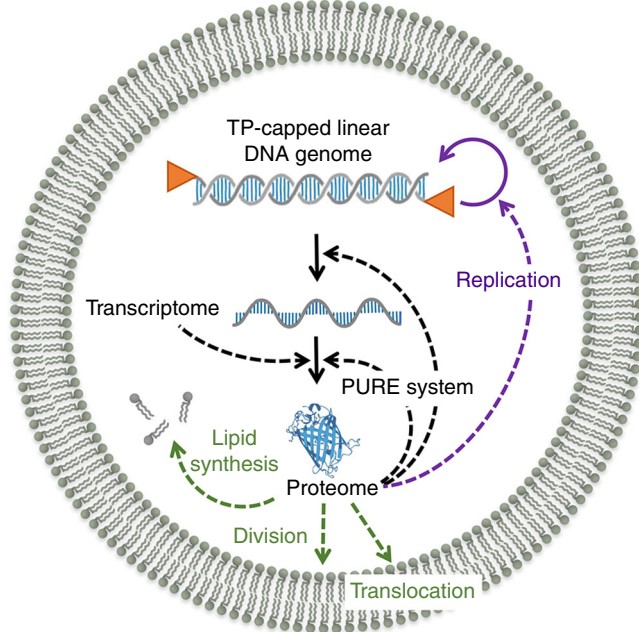

**Fig. 6** A semisynthetic cell with implemented Φ29-based linear DNA replication. A prospective minimal cell, whose chassis is based on the PURE*frex* protein factory encapsulated inside phospholipid vesicles, is represented with its essential functional modules. The transfer of information (black solid lines) from DNA to protein is executed by the PURE system. Dashed lines indicate catalysis reactions. The DNA replication module, whereby the linear genomic DNA is capped with the Φ29 TP protein (triangles) and is replicated by the Φ29 DNA synthesis machinery, has been implemented in this study (purple). Other subsystems include the regeneration of all PURE system components from their genes, the synthesis of phospholipids for the growth of the compartment, the expression of division proteins, and incorporation of transmembrane proteins (channels, transporters) to regulate the molecular diffusion with the external environment, in particular of energy-rich compounds. One challenge to realize a fully functional cell will be to efficiently interface and coordinate the different modules, something that could be fostered by DNA replication through random generation and in vitro selection of favorable phenotypic traits. Gene regulatory circuits (not depicted) could be implemented to orchestrate the expression dynamics of the different modules

**Coupled IVTT and DNA replication**. The cell-free gene expression system PURE*frex* was purchased from GeneFrontier Corporation (Japan). The kit comes in three vials: the enzyme mixture (T7 RNA polymerase, translation factors, energy recycling system, etc.), the buffer (feeding) mixture (amino acids, NTPs, tRNAs, etc.), and the ribosome solution. The PURE*frex* solution for a 20-μL reaction consists of 10 μL buffer solution, 1 μL enzyme solution, 1 μL ribosome solution, 10–150 ng of input DNA template and RNase-free milliQ to fill up the volume. For coupled expression-DNA replication experiments, the standard PURE*frex* reaction mix was supplemented with 2 μL of 200 mM ammonium sulfate (20 mM final concentration) and 0.6 μL of 10 mM dNTP mix (0.3 mM final concentration). Reactions were incubated in a nuclease-free PCR tube (VWR) in a ThermalCycler (C1000 Touch, Biorad) at a default temperature of 30 °C. Specific conditions, e.g., reaction volume, incubation time and temperature, and other supplements (e.g., purified proteins), are indicated when appropriate.

**Amplification of the Φ29 genome**. The Φ29 genome was amplified in the PURE system by in situ expressed proteins, largely as described above. The PURE*frex* reaction mix (30 μL final volume) was supplemented with 140 ng (0.39 nM final concentration) starting amount of the Φ29 genome and 25 ng (0.42 nM final concentration) *oriLR-p2-p3* template. When indicated, additional DNAs were co-expressed: 120 ng (4.7 nM) *oriLR-p6-p5* template, 60 ng (6.4 nM) *p5* or 60 ng (7.2 nM) *p6*. After the reaction, DNA was purified and analysed on alkaline agarose gel as described below.

**Fluorescence labeling and gel imaging of expressed proteins**. The standard PURE*frex* reaction mixture was supplemented with 0.5 μL BODIPY-Lys-tRNA$_{Lys}$ (FluoroTect™ GreenLys, Promega) to fluorescently label translation products at the sites of a lysine residue (10 μL final reaction volume). Around 3 nM of the DNA templates containing the *p2*, *p3*, *p5*, or *p6* gene were separately expressed for 3.5 h at 30 °C. Co-synthesis of the four replication proteins was performed using the *oriLR-p2-p3* and *oriLR-p6-p5* templates starting with 0.7 nM of both constructs, or 0.35 nM of *oriLR-p2-p3* and 4.7 nM *oriLR-p6-p5* DNAs. To verify that protein encoding is retained in the de novo synthesized DNA, 3.2 μL of DNA purified from replication samples (following the protocol for preparation of the TP-capped *oriLR-p2-p3* DNA template, see below) was used as template in a new PURE*frex* reaction supplemented with 0.5 μL Superase (20 U/μL, Ambion). Reactions were carried out for 4–5 h at 30 °C.

For the experiments shown in Fig. 1d, e, samples were treated with RNase A (0.1 mg/mL final concentration) for 30 min at 30 °C to degrade the unreacted labeled tRNA$_{Lys}$, reducing the fluorescence background on the gel. For experiments starting from pre-replicated DNA templates, this step was omitted. Then, 5 μL was mixed with 2× Laemmli Sample buffer and 10 mM DTT (final concentration), denatured for 2.5 min at 65 °C and analyzed on a 12 or 15% SDS polyacrylamide gel electrophoresis (PAGE) gel. Fluorescence detection of the labeled translation products was performed using a fluorescence gel imager (Typhoon, Amersham Biosciences). Visualization of the total proteins (PURE system proteins and purified replication proteins) was realized by Coomassie Brilliant Blue (Promega) after-staining on a GelImager.

**Analysis of DNA on agarose gels**. The protocol to analyze DNA in PURE*frex* reaction samples on neutral agarose gels involves both a protein and an RNA removal step (Supplementary Note 2). These steps are essential for reliable analysis of the DNA band intensities. Samples of 5 μL were supplemented with 0.5 μL of 4 mg/mL RNase A (0.4 mg/mL final) plus 0.5 μL of 5–10 U/μL RNase ONE™ Ribonuclease, and incubated for 30 min at 30 °C. A volume of 3 μL of STOP solution (30 mM EDTA, 0.3% SDS) was added, further supplemented with 0.5 μL of 0.1 mg/mL Proteinase K solution. The solution was incubated for 1 h at room temperature and the samples were stored at 4 °C or –20 °C until further use. All time point samples were treated simultaneously with the RNeasy MinElute Clean-up kit (Qiagen) following the manufacturer's protocol, with a final elution step in 14 μL milliQ. The samples were loaded with a 6× DNA loading buffer (Promega) on a 1.1% agarose gel containing EtBr and were run in TAE buffer.

The sample treatment for DNA analysis on alkaline agarose gels consists only of quenching the reaction with the STOP solution and Proteinase K. Other additions of enzymes need to be performed before this step. When indicated, a lambda exonuclease (0.3 μL from a 1.5 U stock, New England Biolabs) treatment was performed for 30 min at 30 °C before the quenching step in order to digest all DNAs with free 5′-phosphate ends. All PURE*frex* samples containing the Φ29 genome were analyzed on alkaline gels because the RNA column purification does not recover the long genome. Following the same protocol but omitting the column purification step prevents the DNA from running into the gel.

Alkaline conditions hydrolyze RNA and separate DNA strands, allowing to separate unfolded ssDNA according to their respective sizes. The alkaline agarose gels were prepared according to the protocol recommended by Thermo Fisher Scientific (Alkaline agarose gel electrophoresis, 2006). Briefly, a 0.7% agarose gel was prepared in a sodium-chloride buffer (30 mM NaCl, 2 mM EDTA, pH 7.5) and pre-run in the alkaline electrophoresis buffer (30 mM NaOH, 2 mM EDTA) for at least 1 h. A 6× alkaline loading buffer (180 mM NaOH, 6 mM EDTA, 18% Ficoll 400) was added to the samples that were heated for 5 min at 70 °C, then chilled on ice for 3 min prior loading onto the gel. A small fraction (ca. 0.1%) of bromophenol blue was added to the alkaline loading buffer. Gels were run at low voltage (35–45 V) for 4–5 h in the alkaline buffer. Then, the gel was immersed for 30 min in 200 mL of 0.5 M Tris-HCl buffer, pH 7.5, and stained with EtBr.

**Preparation of DNA template capped with TP**. A PURE*frex* reaction mix was supplemented with (final concentrations or masses) 20 mM (NH$_4$)$_2$SO$_4$, 0.3 mM dNTP mix, 7.5 μg purified p5, 2.1 μg purified p6 and 1.3–2.5 nM of the *oriLR-p2-p3* DNA template in a final volume of 20 μL. Control reactions were run in parallel to check the replication and TP-capping efficiency, including a reaction without dNTPs. Samples were incubated for 4–6 h at 30 °C. Then, RNA was removed by addition of 2 μL of 5–10 U/μL RNaseONE™ (Promega) and 2 μL of 4 mg/mL RNaseA (Promega) followed by an incubation step at 30 °C for 40 min. RNase treatment was stopped by addition of 12 μL STOP mix. The control samples were subsequently incubated with 2 μL of 0.1 mg/mL Proteinase K to remove all proteins attached to the DNA. All samples, including the one treated with Proteinase K, were purified with RNaesy mini elute clean-up kit. Uncapped DNA was removed by lambda exonuclease treatment by adding 2 μL 10× lambda exonuclease buffer and 1 μL of 5 U/μL lambda exonuclease (5 U/μL, New England Biolabs) in a final volume of 20 μL. Samples were incubated for 2–3 h at 37 °C. The samples were purified with RNaesy mini elute clean-up kit.

Concentration of the TP-capped *oriLR-p2-p3* DNA was estimated on an agarose gel post-stained with SybrGold (Thermo Fisher Scientific), comparing the band volume of TP-capped DNA to a known concentration of *oriLR-p2-p3* PCR product loaded on the same gel (Supplementary Fig. 12b). The gel was analyzed on a

Typhoon scanner (GE healthcare) with a 488-nm laser and 520 BP emission filter. Band volumes were determined with the ImageQuant TL software program. The amount of TP-capped DNA could slightly be underestimated due to the smear of the bands.

**Coupled IVTTR inside liposomes.** Liposome experiments were adapted from the protocol described in ref. [25]. To prepare lipid-coated beads, a lipid mixture consisting of DOPC (50.8 mol%), DOPE (35.6 mol%), DOPG (11.5 mol%), cardiolipin (2.1 mol%), DSPE-PEG(2000)-biotin (1 mass%) and DHPE-TexasRed (0.5 mass%) for a total mass of 5 mg, was assembled in a 25-mL round-bottom glass flask. All lipids were purchased at Avanti Polar Lipids and dissolved in chloroform, except the DHPE-TexasRed membrane dye that was from Invitrogen. To improve lipid film swelling, 63.5 μmol of rhamnose (Sigma-Aldrich) dissolved in methanol, was supplemented to the lipid mixture. Although rhamnose reduces translation rate and output protein concentration in bulk reactions (Supplementary Fig. 18), it has an overall beneficial role in liposome experiments. Finally, 1.5 g of 212–300-μm glass beads (Sigma-Aldrich) was added to the lipid solution, and the organic solvent was removed by ~2 h of rotary evaporation at 200 mbar, followed by overnight desiccation. The dried lipid-coated beads were stored under argon at −20 °C, and were redesiccated for at least 30 min before use.

A 10-μL PUREfrex reaction was assembled as described above. The solution was supplemented with (all final concentrations) 5 or 10 ng/μL (equiv. 2.5 or 5 nM) oriLR-p2-p3 DNA, 20 μM of purified p5, 8 μM of purified p6, 20 mM of ammonium sulfate, 0.75 U/μL SUPERase (Ambion), and 250 μM of PCR Nucleotide mix (Promega). Alternatively, 1.3 μL or 2.5 μL of purified TP-oriLR-p2-p3 DNA along with 4.2 ng/μL of the p6-p5 construct was used. In the negative controls, the dNTP mix was replaced with an equal volume of milli-Q water. A control experiment for direct visualization of transcription-translation was performed using 7.4 nM of the YFP-spinach construct, whose expression produces fluorescent reporters of the synthesized mRNA and protein[50, 51]. To the 10-μL reaction, ~12 mg of lipid-coated beads was added. Lipid film swelling was performed for 2 h on ice, followed by four freeze-thaw cycles. Using a cut tip to prevent liposome breakage, 2 μL of the liposome solution was added to 5.5 μL of feeding solution, consisting of milli-Q, PUREfrex Solution I (v/v 7:4) and 91 μg/mL proteinase K. The diluted liposome mixture was incubated overnight at 30 °C in an Eppendorf tube.

Next, the liposome solution was transferred to a custom-made imaging chamber. A glass coverslip was functionalized with BSA-biotin (Thermo Fisher Scientific) and Neutravidin (Sigma-Aldrich) for liposome immobilization, as previously reported[50]. Acridine Orange (Sigma-Aldrich) was added at a final concentration of 62.5 μM to visualize DNA. After 30 min of incubation at 37 °C, the liposome sample was imaged with a Nikon A1R laser scanning confocal microscope with a SR Apo TIRF 100× oil immersion objective, using the 458 nm (spinach), 488 nm (acridine orange), 514 nm (YFP) and 561 nm (Texas Red) laser lines with appropriate emission filters.

Microscopy images were analysed with MATLAB (MathWorks). To identify DNA spots, an intensity threshold of (avg + 5 × sdv), using the average and standard deviation of the non-zero pixel values of the acridine orange channel with the texas red channel subtracted, was chosen.

**Data availability**. The authors declare that the main data supporting the findings of this study are available within the article and its Supplementary Information file. Extra data are available from the corresponding author upon request.

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

## Acknowledgements

We thank José M. Lázaro for the purification of the viral proteins and Alicia Soler Canton for performing a protein gel electrophoresis experiment. We also thank Emma Gerritse for assistance with some bulk replication experiments and Anne Doerr for useful discussions about the PURE system. This work was financially supported by the Netherlands Organization for Scientific Research (NWO/OCW) as part of the Frontiers of the Nanoscience Program and through a VIDI grant (project number 723.012.007) to C.D.; M.S. was funded by the Grant BFU2014-52656-P from the Spanish Ministry of Economy and Competitiveness.

## Author contributions

P.v.N. conceived the DNA replication strategy, designed experiments, analysed data, performed bulk experiments and co-wrote the manuscript. I.W. performed bulk experiments, analysed data and edited the manuscript. D.B. performed the liposome experiments, analysed data, and wrote the corresponding experimental section of the manuscript. M.S. and M.M. supervised the preparation of the viral proteins and gave valuable advices for their optimal handling and utilization. C.D. conceived the synthetic cell research, supervised the study, designed experiments, analysed data, and wrote the manuscript. All the authors discussed the results and commented on the manuscript.
