## [Peer Review File · Nature Communications]

Reviewers' comments:

Reviewer #2 (Remarks to the Author):

The authors demonstrated a reconstruction of the process of central dogma based on the cell-free (PURE) system and Φ 29 virus replication system. This study is located in artificial cell or minimal cell research and presents an important evidence showing that replication and expression of genetic information can be implemented within cell-free system. Eventually, the authors have tried to reconstruct this in liposome lumen. The reconstruction of self-reproduction property of life system is one of the hottest topic in this field and, in fact, there are some interesting reports recently from Japanese teams, Fujiwara et al. (2013) and Su'etsugu et al. (2017).

Although the significance of this study is important in the related field, the authors should clearly describe what point of this work has advantage and novelty as compared to these previous reports. The manuscript contains abundant data and many experimental elaborations. However, the reviewer thinks that some experiments are not finished with detailed biochemical analysis and sufficient controls. This may be come from difficulty of the designed experimental composition.

Followings are major and minor comments.

Fig.1E: Cell-free expression of each single gene and its yield was estimated from the data of SFig.3. However, the data of SFig.3 needs more information, for example what are the protein concentrations of each protein of three different mixtures, how these concentrations were defined first. The standard curve also should be supplied.

Figure 1 is also missing the quantitative data of co-synthesized each protein, although most of the following experiments are performing co-synthesis of p2 and p3 proteins.

Fig. 2B: What is the extra band marked as the red asterisk? The band intensities of replicated Φ 29 genome looks same after 1 and 4 hours reactions, in the absence of p5/p6. This may indicate that the extra band was produced after the completion of Φ 29 genome replication. Additionally, this did not appear in all purified proteins were used (SFig. 7). The authors should explain more about this extra band.

Fig. 3B: About the 4th lane from the left, why there is a band at the position of oriLR-p2-p3, although p2-p3 template DNA was used?

C: The successfully replicated and capped oriLR-p2-p3 was detected even after the endonuclease treatment, especially in the case of the purified proteins (+) and dNTPs. The amount of the appeared oriLR-p2-p3 should be dependent of the amount of synthesized P3. Thus, authors should show the quantitative consistency between the synthesized p3 and the replicated and remained oriLR-p2-p3. Additionally, in the control data of the p2-p3 case, some faint bands were appeared when the purified proteins and dNTPs are (+). Why this was happened? Accuracy of the replication of the cell-free synthesized DNAP should be shown by comparing DNA sequence data of the replicated and original.

D: De novo p2 and p3 were synthesized from the replicated oriLR-p2-p3 as shown at dNTPs (+) or both the purified proteins and dNTPs (+). In this result, p3 protein was synthesized in preference to p2. Why? Is it depending on the order of genes on the genome? Authors described in the main text as "... Sufficient to unambiguously produce p2 and p3 proteins." But p2 product in dNTPs solo (+) cannot be said unambiguously from this gel data. This description should be reconsidered.

As overall of this result, quantitative description is missing, i.e., "x nM original DNA" produced "a nM proteins" that produced "y nM TPed-replicated DNA", which finally synthesized "b nM proteins". This quantitative summarization is important to understand the reaction flow of this study.

Fig. 4B: This data shows effect of TP on the original template DNA. The effect of TP is significant,

but many intermediate products were also appeared. On the other hand, intensity of the extra band above TP-oriLR-p2-p3 was reduced as compared to Fig.3B. How is this explained? Also, the many intermediate bands were disappeared when p6-p5 was co-expressed. Why?

C: A large amount of the replicated TP-oriLR-p2-p3 was degraded by nuclease at lane 3 and 5. From this data, it is shown that co-expressed p5 and p6 enhanced replication but inhibited the binding of TP at the 5' end of DNA. What mechanism leads such result?

D: Almost same p2 and p3 were synthesized in all three samples, although the sample of TP-oriLR-p2-p3 and purified p5 has been most replicated at the previous step. How the authors explain this?

Fig. 5B: It is rather difficult to proof that the replication was performed in liposomes only from this data. The fluorescent signal caused by the DNA replication within liposomes should be measured by a fluorescence spectrophotometer. The reviewer wonders why DNA dye is not distributing within liposome lumen. This may indicate that the replicated DNA forms aggregate? If so, same stop fluorescence can be observed when other DNA dye, such as SYBR Gold or Green, was used? Although authors explain green spots were appeared in the + dNTPs case, spots are also observed in -dNTPs, where most of spots are quasi lipidic spots. The four liposomes, which contain only green spot, were selected and analyzed. But those frequency looks very low. In any case, more control experiments are needed.

C: line plot data of -dNTPs liposomes are also required.

Fig.6: The reviewer thinks that an energy acquisition of semisynthetic cell should also be described in this figure.

P23, line15: Please explain why ammonium sulfate was additionally added in the reaction mixture.

P25, line 3: Please show if rhamnose affects to the protein synthesis in PURE system or not.

Reference: It is weird that the paper of Su'etsugu et al. (2017 NAR), which is also focusing gene replication in reconstructed manner, is not cited.

Reference 32: (2013) is missing.

Supplementary Information

P4, line11: DNA concentration should be written as nM.

SFig.4: Was the same concentration of DNAP analyzed in B and C experiment?

C: Why Linear RAC product is smearing in -fluoro-dUTP at 60 min?

SFig.5C: Was the same concentration of DNAP analyzed in produced and purified DNAP?

SFig.9: The experiment flow of this study is very complicate. To make this understand easily, a schematic of workflow should be added in this figure.

DNA concentrations should be written as nM.

SFig.10: What does the smear on dNTP (+) lane indicate? What is the band which appeared above 1000kbp on dNTP (+) lane, and why this appeared?

DNA concentrations should be written as nM.

SFig.10: Please explain the working mechanism that the purified p5 increased p3 productivity and the purified p5 and p6 increased p2 productivity.

SFig.12B: Authors estimated the amount of TP-oriLR-p2-p3 (dNTPs+) from the band intensity of reference DNAs. However, the band intensity of the product is much bigger than references. Thus, the obtained quantification value cannot be reliable.

D: Why there are p2 and p3 products in dNTPs (-) control?

E: Is this sample same as that analyzed in B? If so, why smearing and intermediate pattern is different?

SFig. 13: Add the quantification data of the replication in capped and non-capped template DNA.

SFig. 14: More control data are required, for example, TP-oriLR-p2-p3 DNA (-), p5/p6 (-), etc.

Reviewer #3 (Remarks to the Author):

In this study titled "Self-replication of DNA by its encoded proteins in liposome-based synthetic cells", the authors performed DNA self-replication by in vitro-translated phi29 proteins. They successfully translate the proteins, p2, p3, p5, and p6, of the phi29 proteins in the active forms and performed replication of the oriLR-DNA by these proteins. They also showed that the replicated DNA can be a template for the next round of gene expression and replications. Moreover, they performed the gene-expression-coupled DNA replication in liposomes. To date, several groups have tried to constitute translation-coupled DNA self-replication system in vitro, such as phi29 DNAP-cre system and E.coli genome replication system, while recursive DNA replication has not been realized. This study, demonstrating at least three round successive replication, provides a significant advance for the constitution of in vitro DNA replication system. The manuscript was written precisely. Most of the claims the authors raised are convincing with a large amount of supportive data. Therefore, I believe that this study is suitable for publication if the authors appropriately respond the minor points as described below.

Minor points

1. In Figure 1A, I could not understand the difference between "oriLR-DNA" and "(oriLR-)DNA". Does the "-" in the latter mean "minus" , not "hyphen"?

2. Figure 1C. Why does not the phi29 genome contain any genes?

3. Why did the authors use RNeasy column for purification of DNA instead of QIAquick column for DNA?

4. Fig 4E is ununderstantable because the third replication was not performed in Figure 4A-D. Fig. 4E should be in the Supplementary Fig. 12, where the replication continued until third round.

5. In page 16, the authors wrote, "Darwinian evolution requires colocalization of the genotypic and phenotypic components", which is incorrect statement. Evolution of a good template (e.g., a template DNA that can be replicated faster) can evolve without any compartments. For the evolution of encoded proteins, a linkage between genetic molecule and phenotypic molecule (e.g., by compartmentalization) are required.

6. Figure 5. It would be helpful to provide the information about how many DNA were in one liposome on average.

7. Figure 5. I think that the use of acridine orange, which also bind to RNAs in PURE system, cause a high back ground fluorescence in liposomes, which mask the fluorescence of the replicated DNA. I don't think it is necessary for this paper, but it would be better to use other dyes that are more specific to dsDNA, such as SYBR green I.

8. In the legend of Figure 5, the author wrote, "Due to the stochastic nature of the encapsulation". This sounds strange because for this size of liposome, stochastic nature would appear only for the

components at less than 1 nM. All PURE system components and probably the template DNA are at much higher concentrations.

9. In the legend of Figure 5, the author wrote, "acridine orange can stain the liposome membrane". Other possibilities are that the RNAs in PURE system (tRNA and mRNA) were stained by acridine orange and attached to the membrane or the membrane fluorescence by Texas-red might leaked. The authors should check these possibilities or just mentioned other possibilities.

10. Supplemental Information page 7. " β Q replicase" should be "Q β replicase".

Response to Referees letter

Manuscript NCOMMS-17-28406-T entitled: "Self-replication of DNA by its encoded proteins in liposome-based synthetic cells"

Note to the reader about the color coding: The Reviewers' comments are in black type, our reply is in blue type and text changes have been highlighted in red type, both in the present document and in the revised manuscript.

Reviewer #2 (Remarks to the Author):

We appreciate the efforts from the reviewer to provide extensive comments and suggestions that helped us improve the clarity of the manuscript.

The authors demonstrated a reconstruction of the process of central dogma based on the cell-free (PURE) system and Φ 29 virus replication system. This study is located in artificial cell or minimal cell research and presents an important evidence showing that replication and expression of genetic information can be implemented within cell-free system.

Thank you for the positive statement. It is indeed our main finding.

Eventually, the authors have tried to reconstruct this in liposome lumen. The reconstruction of self-reproduction property of life system is one of the hottest topic in this field and, in fact, there are some interesting reports recently from Japanese teams, Fujiwara et al. (2013) and Su'etsugu et al. (2017).

The work of Fujiwara et al. (2013) is highly relevant and was already cited as reference [32]. We were not aware of the more recent study by Su'etsugu et al. (2017). The authors reconstituted the *E. coli* replication machinery with 14 purified proteins. Even though the mode of replication and the overall approach differ from our work, we agree that this study is relevant and decided to refer to it on page 3 as new reference [31]. Noteworthy, the results presented by Su'etsugu et al. do not lessen the innovative aspects and significance of our work.

New ref. 31:

Su'etsugu, M., Takada, H., Katayama, T. & Tsujimoto, H. Exponential propagation of large circular DNA by reconstitution of a chromosome-replication cycle. *Nucleic Acids Res.* **45**, 11525–11534 (2017).

Although the significance of this study is important in the related field, the authors should clearly describe what point of this work has advantage and novelty as compared to these previous reports.

Thank you for stating that the significance of our study is high in this field.

In the introduction, the part starting from "In parallel, several isothermal DNA replication machineries..." up to "... is necessary to regenerate the parental DNA." was precisely

dedicated to comparing existing approaches with ours. In addition, on page 7 of the Supplementary Information, we devoted a paragraph that further explains the difference between the Φ 29-based mechanism described here and alternative approaches. Finally, the overall significance and novelty were emphasized in the first paragraph of the Discussion section. Although it was already stated in the Introduction, the minimalist nature of the Φ 29 replication machinery compared to the bacterial systems (including the studies from the Japanese groups) is one advantage that needs to be more clearly stated. Therefore, we inserted the following sentence at the end of the Discussion: “The fact that the protein-primed DNA replication mechanism of phage Φ 29 is composed of only four proteins (vs. >12 for the bacterial systems [31,32]) is an advantage because its expression involves low usage of resources.”

Hence, to our opinion, the motivation and advantages of our approach are now completely covered. Due to limited space in the main text, we prefer to not elaborate more extensively.

The manuscript contains abundant data and many experimental elaborations. However, the reviewer thinks that some experiments are not finished with detailed biochemical analysis and sufficient controls. This may be come from difficulty of the designed experimental composition.

Below, we address point by point the specific reviewer’s concerns about the biochemical analysis.

Followings are major and minor comments.

Fig.1E: Cell-free expression of each single gene and its yield was estimated from the data of SFig.3. However, the data of SFig.3 needs more information, for example what are the protein concentrations of each protein of three different mixtures, how these concentrations were defined first. The standard curve also should be supplied.

We added the standard curves for all four proteins as a new panel in Supplementary Fig. 3B (see graph below) and included the following sentences in the legend. “Standard curves representing the band intensity values for the purified proteins of known concentrations (circle symbols) were measured from the gel shown in A). Linear fits were calculated and displayed as dashed lines. The band intensity of the expressed protein (square symbols) was appended on the linear fit. The corresponding mass was calculated and converted into concentration. Linearity of the calibration curves down to 25 ng protein was confirmed in other experiments.”

The preparation and quantitation of the purified proteins was done as described in references [1-3]. This was already mentioned on page 1 of the Supplementary Information. We agree to add the following description in the same paragraph: “Quantification was done by the Lowry method using Coomassie staining of SDS-PAGE gels with appropriate standards followed by densitometry of the bands. In the case of the DNA polymerase, absolute concentration was also assessed by measuring polymerization activity against known specific activity. Stock concentrations and storage buffers are: ...”

Figure 1 is also missing the quantitative data of co-synthesized each protein, although most of the following experiments are performing co-synthesis of p2 and p3 proteins.

In Fig. 1D,E, co-expression of all four proteins is shown. The resulting bands on CBB (D, lanes 6 and 7) are too weak for quantification. The corresponding bands are clearly visible on the GreenLys gel (E, lanes 6 and 7). However, GreenLys is not a quantitative method and cannot be used to reliably infer the concentration of synthesized proteins. The reason lies in the fact that the occurrence of lysine residues varies from one protein to the other, and that the insertion efficiency of a fluorescent lysine (that competes with the unlabelled lysine) at a given position is an unknown parameter that varies in the course of gene expression. To inform the reader about the limitations of using GreenLys for protein quantitation, we added the following sentence in Supplementary Materials and Methods on page 3: “Co-translational insertion of fluorescently-labelled lysine can be used to visualize synthesized proteins by PAGE, even when their concentrations are lower than the detection limit with CBB staining (Fig. 1E). However, GreenLys is not a reliable method for protein quantitation because the insertion efficiency of a fluorescent lysine (in competition with unlabelled lysine) at a given position is largely unknown and it varies in the course of gene expression. Moreover, the band intensity depends on the occurrence of Lys residues in the primary sequence, which can differ from one protein to another. Therefore, we chose CBB staining in combination with purified protein standards to estimate the concentrations of synthesized proteins (Supplementary Fig. 3).”

As suggested by the reviewer, we performed additional gel electrophoresis experiments to quantify the amounts of p2 and p3 proteins synthesized from the *oriLR-p2-p3* construct under conditions used in following IVTTR assays. Purified p2 and p3 protein standards were used at four different concentrations for calibration and samples with the expressed *oriLR-p2-p3* construct were loaded on the same gel. The results are reported in the new Supplementary Fig. 3C,D (see Figure below, including the accompanying legend).

Legend: “(C) The *oriLR-p2-p3* construct (2.5 nM) was expressed and the amount of synthesized p2 and p3 proteins was estimated by PAGE. Purified p2 and p3 were loaded on the same gel at four to five different concentrations for calibration. (D) Standard curves and concentrations of expressed p2 and p3 were determined as described in (B). Under these conditions, ~100 nM of p2 (with or without dNTPs), and ~700 nM (-dNTPs) or ~180 nM (+dNTPs) of p3 were measured. The lower concentration of p3 detected in the presence of dNTPs may be assigned to its depletion by DNA linkage upon replication initiation. Although these concentrations are significantly reduced compared to IVTT from single-gene DNA templates (B), they exceed the amounts that are sufficient for *in vitro* amplification of heterologous DNA, i.e. 16 nM for p2 and 65 nM for p3 [12].”

Importantly, although we can only provide an estimation of synthesized p2 and p3 concentrations from the *oriLR-p2-p3* templates, we know from the replication assays that their individual amounts are sufficient to amplify DNA. In fact, only 16 nM of p2 and 65 nM of p3 are sufficient for *in vitro* amplification of heterologous DNA (see ref [12]). As also shown in Supplementary Fig. 7, the $\Phi 29$ genome was efficiently replicated with 32 nM p2 and 65 nM p3. The concentrations of p2 and p3 produced from the *oriLR-p2-p3* construct exceed these values.

Actually, future works should aim at reducing the concentration of synthesized p2 and p3 to limit production of full-length DNA (see comment on page 6 of this document).

In the legend of Fig. 1D, we added the amounts of purified proteins in lanes 8 and 9: “Amounts of purified proteins: 180 ng p2, 2 μ g p5, 180 ng p3 and 2 μ g p6.”

Fig. 2B: What is the extra band marked as the red asterisk? The band intensities of replicated Φ 29 genome looks same after 1 and 4 hours reactions, in the absence of p5/p6. This may indicate that the extra band was produced after the completion of Φ 29 genome replication. Additionally, this did not appear in all purified proteins were used (SFig. 7). The authors should explain more about this extra band.

As stated in the figure caption, “the red asterisk indicates the upper band of the Φ 29 genome” and we referred the reader to Supplementary Fig. 7 and 8, where we give a possible explanation for this extra band, specifically: “... probably due to the tendency of TP to aggregate in the gel or to the formation of genome dimers.” Indeed, this band does not appear in the presence of purified (Supplementary Fig. 7) or synthesized (Fig.2B, right panel) p5/p6 proteins. To be more complete about this observation, we added the following sentences in the legend of Supplementary Fig. 7: “The upper band (marked with an asterisk) is only visible when high amounts of Φ 29 genome are loaded on gel and it does not appear in the presence of purified or synthesized (Fig.2B, right panel) p5 and p6 proteins. A possible cause is the tendency of TP (that resisted protease digestion) to aggregate in the gel or to the formation of genome dimers.”, and the following one in the legend of Supplementary Fig. 8: “Note the presence of an upper band (indicated by an asterisk) in both gels as already reported in Fig. 2B and Supplementary Fig. 7. After heat denaturation of the Φ 29 genome, only one band of ssDNA is visible. This indicates that the upper band corresponds to the Φ 29 genome present in a different state or folded structure.”

Fig. 3B: About the 4th lane from the left, why there is a band at the position of *oriLR-p2-p3*, although p2-p3 template DNA was used?

Actually the upper band is slightly lower than that of *oriLR-p2-p3* and we are sure it does not contain the *oriLR* sequences. Appearance of an upper band is a recurrent observation across the different DNA templates tested, namely the Φ 29 genome, *p2-p3* and *oriLR-p2-p3*. As can be seen on the same gel, amplification of *oriLR-p2-p3* in the presence of purified p5 and p6 also gives rise to an upper band. Although we do not know the exact nature of this DNA and the underlying mechanism that produces it, we suppose it originates from some interwoven dsDNA and ssDNA due to uncompleted replication, from DNA aggregation via TP that resisted protease digestion, or from a distinct long-sized replication product whose nature remains to be elucidated.

We agree that this point deserves more attention and we added the following text in the legend of Supplementary Fig.9A (notified in Fig.3B): “Efficient replication mediated by the purified p5 and p6 proteins also leads to a band above the full-length product (see also Fig. 3B). Although we do not know the exact nature of this DNA product and the mechanism that generates it, we presume it originates from some interwoven dsDNA and ssDNA due to uncompleted replication, from DNA aggregation via TP that resisted protease digestion or from a distinct long-sized replication product whose nature remains to be elucidated.”

C: The successfully replicated and capped *oriLR-p2-p3* was detected even after the endonuclease treatment, especially in the case of the purified proteins (+) and dNTPs. The

amount of the appeared oriLR-p2-p3 should be dependent of the amount of synthesized P3. Thus, authors should show the quantitative consistency between the synthesized p3 and the replicated and remained oriLR-p2-p3.

The reviewer assumes that the amount of replicated DNA depends on the amount of synthesized p3 and that their relationship can be predicted. Although in theory every newly synthesized DNA strand is linked at its 5' end to a TP (p3), it does not necessarily imply that p3 concentration is the limiting factor controlling replication efficiency. In fact, it is known for instance that low concentrations of p3 and p2 favour faithful replication (as stated in the legend of Supplementary Fig. 7). Moreover, the proteins p5 and p6 regulate the production of full-length products versus short replicons, a process that is not directly dependent on the concentration of p3. Another reason supporting that there exists no direct correlation between the concentration of p3 and the amount of full-length replication product is the possibility that the replication machinery may interfere with the transcribing RNA polymerase.

Nonetheless, one can expect that the concentration of p3 sets an upper limit for the level of amplified DNA. The first generation duplicated DNA is linked with one TP, while subsequently amplified DNA contains two TP's. Starting from 2.5 nM *oriLR-p2-p3* template, assuming 700 nM of p3 (Supplementary Fig. 3) with an excess amount of dNTPs, an estimation of the maximum amplification yield of full-length DNA is ~140-fold.

We added the following sentence at the end of the legend of Supplementary Fig. 3:

“Moreover, the upper limit of amplified DNA concentration with an excess of dNTPs is imposed by the concentration of p3, here ~700 nM, which is sufficient to amplify DNA by a factor ~140. Future works should aim at reducing the amount of synthesized p3 to limit replication of DNA to a level that is adequate for functional transmission of the genetic information upon division.”

Additionally, in the control data of the p2-p3 case, some faint bands were appeared when the purified proteins and dNTPs are (+). Why this was happened? Accuracy of the replication of the cell-free synthesized DNAP should be shown by comparing DNA sequence data of the replicated and original.

As for the *oriLR-p2-p3* construct, the lower bands observed when *p2-p3* is replicated can be attributed to the formation of short replicons, as already discussed. As some of them remain after lambda exonuclease treatment, they must be linked to TP.

The accuracy of DNA replication is demonstrated by the fact that two successive rounds of IVTTR and an extra IVTT reaction have been achieved (Fig. 4D and additional text in the legend). These results demonstrate that, at every step of the recursive cycle, a sufficient fraction of the reaction products (DNA, mRNA and proteins) is functional. As reported on page 7 of the Supplementary Information, section “Validation of the activity of the synthesized Φ 29 DNA polymerase”, we performed various assays to confirm the different properties of the expressed DNAP: strand-displacement, polymerization, TP-primed replication initiation, incorporation of non-natural nucleotides, with systematic comparison with the purified enzyme. All activities were validated with no significant differences with the purified DNAP. Therefore, we do not see any reasons to believe that the inherent fidelity

of the expressed p2 will be altered compared to that of the purified enzyme, at least in the conditions tested here. It is true that DNA sequencing would give insights on replication accuracy at the single nucleotide level. However, we believe that such an investigation goes beyond the scope of the present study.

D: De novo p2 and p3 were synthesized from the replicated *oriLR-p2-p3* as shown at dNTPs (+) or both the purified proteins and dNTPs (+). In this result, p3 protein was synthesized in preference to p2. Why? Is it depending on the order of genes on the genome?

In that protein gel (Fig. 3D), the band of p3 is more intense than the one of p2. However, it does not mean that p3 is synthesized at higher amount than p2. The GreenLys labelling is not quantitative; the band intensity depends for instance on the occurrence of Lys residues in the primary sequence, which is different for the two proteins (64 for p2 and 27 for p3). In Fig. 1D,E where the individual genes have been expressed, the p2 and p3 band intensities in the GreenLys and CBB-stained gels can be compared. Looking at Fig. 1E (lanes 2 and 3), the bands of p2 and p3 look similar, whereas in Fig. 1D (CBB) the p2 band is clearly more intense, demonstrating that GreenLys is not a reliable method for protein quantitation.

Starting from the *oriLR-p2-p3* construct in the condition mentioned by the reviewer, it is true that the p3 protein band is more intense than that for p2 (also observed in Supplementary Fig. 9B and Fig. 11, including with the *p2-p3* template). However, on the basis of the above considerations, we cannot conclude from GreenLys inspection that p3 is produced at higher yield than p2. To make the reader fully aware about the limitations of the GreenLys labelling strategy to infer protein quantitation, we added the following text in the Supplementary Materials and Methods (SI page 3): “Co-translational insertion of fluorescently-labelled lysine (GreenLys) can be used to visualize synthesized proteins by PAGE, even when present at concentrations lower than the detection limit with CBB staining (Fig. 1E). However, GreenLys is not a reliable method for protein quantitation because the insertion efficiency of a fluorescent lysine (in competition with unlabelled lysine) at a given position is largely unknown and it varies in the course of gene expression. Moreover, the band intensity depends on the occurrence of Lys residues in the primary sequence, which can differ from one protein to another. Therefore, we chose CBB staining in combination with purified protein standards to estimate the concentrations of synthesized proteins.”

In dual gene constructs like the *p2-p3* and *oriLR-p2-p3* templates, it is indeed possible that the order of genes influences transcription efficiency. As can be seen in Supplementary Fig. 2B (lane 4), more mRNA is produced from the *p3* gene than from *p2* although they are both under control of the same promoter. A possible explanation is that the occurrence of transcription initiation of the second gene is higher because its promoter is in close proximity from the upstream terminator of the first gene. As a consequence, the dropped off RNA polymerase could rebind to the second promoter increasing the probability of complex formation compared to that for the first promoter. In general, a larger amount of mRNA does not necessarily imply that the concentration of the encoded protein is higher since translation is a complex and nonlinear process. The estimated concentrations of p2 and p3 expressed from the *oriLR-p2-p3* DNA are 100 nM and 700 nM (–dNTPs), respectively. Therefore, in

this particular example, expression of the second gene gives rise to a higher protein concentration. We added the following text in the legend of Supplementary Fig. 2B: “We noticed that transcription of the *oriLR-p2-p3* template leads to more mRNA from the *p3* gene than from *p2* although they are both under control of the same promoter (lane 4). A possible explanation is that the occurrence of transcription initiation of the second gene (*p3*) is higher because its promoter is in close proximity from the upstream terminator of the first gene (*p2*). As a consequence, the dropped off RNA polymerase could rebind to the second promoter increasing the probability of complex formation compared to that for the first promoter. Transcription assays with dual-gene constructs with different spacing between the two transcription units should be conducted to confirm this hypothesis.”

Authors described in the main text as “... Sufficient to unambiguously produce p2 and p3 proteins.” But p2 product in dNTPs solo (+) cannot be said unambiguously from this gel data. This description should be reconsidered.

Our statement is valid. It is true that the p2 band is faint but it is clearly present. In a repeat experiment shown in Supplementary Fig. 11 (and mentioned in the legend of Fig. 3), the p2 band is more visible.

As overall of this result, quantitative description is missing, i.e., “x nM original DNA” produced “a nM proteins” that produced “y nM TPed-replicated DNA”, which finally synthesized “b nM proteins”. This quantitative summarization is important to understand the reaction flow of this study.

We added the following sentence in the caption of Fig. 3B:

“In each IVTTR reaction triggered by the expression of the *oriLR-p2-p3* DNA construct, 2.5 nM of template produced about 100 nM of p2 and 700 nM of p3 proteins (as estimated in Supplementary Fig. 3), which were able to generate ~50 nM of full-length DNA product when the reaction was supplemented with purified p5 and p6.”

Final estimation of protein concentration from the second IVTT is difficult to assess with Greenlys for the reasons explained above. However, we know that the amount of input de novo synthesized DNA produces p2 and p3 protein bands of similar intensity as that measured when starting with 2.5 nM purified DNA (control with PCR product). We included a new sentence in the legend of Fig. 3D: “Expression of DNA that resulted from an IVTTR in the presence of purified p5 and p6 proteins led to fluorescent p2 and p3 protein bands of similar intensity as that measured when starting with 2.5 nM purified DNA (control with PCR product).”

Fig. 4B: This data shows effect of TP on the original template DNA. The effect of TP is significant, but many intermediate products were also appeared. On the other hand, intensity of the extra band above TP-*oriLR-p2-p3* was reduced as compared to Fig.3B. How is this explained?

Indeed, a large amount of intermediate products is generated, as we already commented in the main text. In Fig.4B, expression with purified p5 and p6 was not tested. Therefore, it

should only be compared with the lane 3 of Fig. 3B, where no extra upper band was observed.

Also, the many intermediate bands were disappeared when p6-p5 was co-expressed. Why? We already described the role of p5 and p6 in the legend of Fig.1 with appropriate references, as well as the possible nature of the replicons. Nonetheless, to be clearer about the possible molecular origin of this experimental observation, we added on page 13: “We interpret the reduction of side products as a manifestation of the functionality of the synthesized p5 and p6 proteins”.

C: A large amount of the replicated TP-oriLR-p2-p3 was degraded by nuclease at lane 3 and 5.

No, the loaded amount of DNA was less than in B. We already indicated in the figure legend that some DNA was not visible “due to dilution during the cleaning/purification steps.” To avoid confusion, we modified the sentence as: “Note that the overall amount of DNA on the gel is reduced (to the extent that the band corresponding to the input TP-oriLR-p2-p3 DNA in the –dNTPs control sample is no longer visible) after nuclease treatment due to dilution during the cleaning/purification steps.”

From this data, it is shown that co-expressed p5 and p6 enhanced replication but inhibited the binding of TP at the 5' end of DNA. What mechanism leads such result?

No, the data do not suggest that p5 and p6 inhibit TP binding. Co-expression of p5 and p6 increases the fraction of full-length replication product by stimulating initiation events that lead to faithful amplification of the substrate.

D: Almost same p2 and p3 were synthesized in all three samples, although the sample of TP-oriLR-p2-p3 and purified p5 has been most replicated at the previous step. How the authors explain this?

As discussed above, GreenLys labelling cannot provide quantitative insights on protein concentrations. The new clarifications in the revised manuscript should dissipate the confusion. On the other hand, protein yield does not linearly correlate with the amount of input DNA. In other words, the amounts of replicated DNA in the three samples might be different (yet high enough to saturate the translation machinery), whilst the actual concentrations of synthesized proteins are similar. Another factor influencing protein yield is the ‘quality’ of the synthesized DNA.

Fig. 5B: It is rather difficult to proof that the replication was performed in liposomes only from this data. The fluorescent signal caused by the DNA replication within liposomes should be measured by a fluorescence spectrophotometer.

We disagree. The observed bright dots in the +dNTPs sample unequivocally result from DNA amplification.

Though we have not performed spectrometry measurements of DNA signal in liposomes, we anticipate that the increase of fluorescence signal upon DNA replication will be far too low for accurate detection. Moreover, the spatial information will be lost. In addition, ensemble

measurements will not provide insights on the number/fraction of liposomes with replicated DNA. Concluding, we gather that direct visualization at the single-vesicle level is the most relevant approach to demonstrate DNA replication inside liposomes.

The reviewer wonders why DNA dye is not distributing within liposome lumen. This may indicate that the replicated DNA forms aggregate?

Correct. As stated in the Discussion and Outlook on page 7: "... amplification of the *oriLR-p2-p3* DNA inside liposome results in punctuated accumulation of the acridine orange dye, indicating DNA aggregation." In the next sentence we also gave a possible cause for aggregation.

If so, same stop fluorescence can be observed when other DNA dye, such as SYBR Gold or Green, was used?

We don't understand what the reviewer means by 'stop fluorescence'. We motivated the use of acridine orange over alternative DNA dyes in the Supplementary Information on page 7. In the revised manuscript, we included a new reference ([45]), in which the formation of similar fluorescent spots was observed in water-in-oil emulsion droplets upon DNA amplification. In this study, the authors used the SYBR Green I dye, suggesting that the process of DNA aggregation is not dependent on the dye.

The following sentence was included on page 21: "Alternatively, accumulation of inorganic phosphate (despite the presence of a pyrophosphatase in the PURE system) during transcription and replication combined with the presence of magnesium ions may trigger the formation of DNA aggregates [45]."

New ref. 45:

Galinis, R., Stonyte, G., Kiseliovas, V., Zilionis, R., Studer, S., Hilvert, D., Janulaitis, A., & Mazutis, L. DNA nanoparticles for improved protein synthesis in vitro. *Angew. Chem.* **128**, 3172–3175 (2016).

Although authors explain green spots were appeared in the + dNTPs case, spots are also observed in –dNTPs, where most of spots are quasi lipidic spots.

Indeed, in the –dNTPs control samples, green spots are also observed. However, they colocalize with the membrane dye, and their occurrence is way lower than in the replication samples. These points were already discussed in the legend of Fig. 5.

The four liposomes, which contain only green spot, were selected and analyzed. But those frequency looks very low. In any case, more control experiments are needed.

This is not correct. We selected four liposomes that exhibited green spots for further intensity profile analysis (shown in panel C), out of many more in the same field of view. We copy below Figure 5B, where we appended solid arrows that point toward liposomes exhibiting such a green spot in their lumen. We did three experimental repeats and counted more than 1,200 vesicles with a clear spot. Quantitative insights were already provided in the caption of Fig. 5. We are confident that the current data fully support our main claims that IVTTR occurs in liposomes and that the amplified DNA forms aggregates.

C: line plot data of $-dNTPs$ liposomes are also required.

We modified Fig.5 to also show line intensity profiles for the $-dNTPs$ sample (only the line intensity profiles are shown in the figure below, not the fluorescence images with the corresponding framed liposomes, see new Fig. 5 in the revised manuscript). We deliberately chose liposomes that exhibited green spots to show their co-localization with the membrane dye, demonstrating that they are of different nature than the selected dots in the $+dNTPs$ image.

The legend of Fig. 5 was modified accordingly: “(C) Line intensity profiles from eight liposomes framed in (B). In the images from the $-dNTPs$ sample, we deliberately chose liposomes exhibiting green spots to show that they colocalize with the membrane dye, demonstrating that they are of different nature than those triggered by DNA replication.”

Fig.6: The reviewer thinks that an energy acquisition of semisynthetic cell should also be described in this figure.

We would like to keep the figure as it is and to not add extra information, even if energy conversion is certainly relevant. Actually, energy considerations were implicit in ‘PURE system’ as it contains an energy regeneration module and in ‘translocation’ for the uptake of energy-rich nutrients. To be more explicit, we added in the legend of Fig. 6: “in particular of energy-rich compounds”.

P23, line15: Please explain why ammonium sulfate was additionally added in the reaction mixture.

Thank you for pointing this out. We added the following sentence in the legend of Fig. 1B: “The TP and DNAP form a 1:1 complex in solution, whose stable assembly is stimulated by the ammonium ion [37].”

New ref. 37:

Blanco, L., Prieto, I., Gutiérrez, J., Bernad, A., Lázaro, J. M., Hermoso, J. M. & Salas, M. Effect of NH₄⁺ ions on phi 29 DNA-protein p3 replication: formation of a complex between the terminal protein and the DNA polymerase. *J. Virol.* **61**, 3983–3991 (1987).

P25, line 3: Please show if rhamnose affects to the protein synthesis in PURE system or not. We conducted a new experiment that we included as Supplementary Fig. 18. The accompanying legend is: “Effect of rhamnose on cell-free gene expression. We used the *T7p-eYFPco-LL-Spinach-T7t* dsDNA construct described in Supplementary Fig. 5 (and in its corresponding experimental section) as a template for a gene expression experiment in cuvette. In one sample, 40 mM rhamnose was added, a concentration similar to that expected in the liposome experiments. An even higher concentration was also tested. The kinetics of mRNA and protein production were simultaneously monitored for all three conditions and the time traces are displayed. The results show that rhamnose reduces the yield of synthesized protein by about 40%, whereas transcription is largely unaltered. Despite its negative influence on translation, rhamnose is a necessary solute to assist formation of large unilamellar vesicles. Further investigations to titrate rhamnose concentration are required to see if better compromising conditions can be found.”

Reference: It is weird that the paper of Su’etsugu et al. (2017 NAR), which is also focusing gene replication in reconstructed manner, is not cited.

This paper is cited in the revised version as new ref. 31. The paper was published only on September 28 this year and we honestly were not aware of it. See also our comments above.

Reference 32: (2013) is missing.

Thank you, it is corrected.

Supplementary Information

P4, line11: DNA concentration should be written as nM.

It is common practice in the specialized literature to report the mass of DNA template as it is also a parameter that influences amplification yield. Whenever relevant we provided the mass, but we systematically indicated also the DNA volumic concentration in nM in the revised manuscript and its Supplementary material.

We included the concentration of DNA in nM at the indicated location.

SFig.4: Was the same concentration of DNAP analyzed in B and C experiment?

In B, 20 units of the DNAP from New England Biolabs was used. The corresponding exact concentration is not directly provided by the supplier but an estimation is 25 nM. In C, p2 was synthesized from 3 nM of gene, and its estimated absolute concentration is 1 μ M. The fraction of active enzymes is unknown, though. The purpose of these experiments was not to draw a comparative quantitative analysis between purified and synthesized DNAP, but to show qualitatively that the synthesized polymerase exhibits strand-displacement activity.

C: Why Linear RAC product is smearing in –fluoro-dUTP at 60 min?

Smearing is probably due to overloading of the gel in this particular condition.

SFig.5C: Was the same concentration of DNAP analyzed in produced and purified DNAP?

5 units of the purified DNAP from NEB was used, corresponding to approximately 6 nM. 90 ng of the p2 gene was used, resulting in about 1 μ M total protein. Again, the goal is not to quantitatively compare the RCA reaction efficiency by the purified and expressed DNAPs, see comment above.

SFig.9: The experiment flow of this study is very complicate. To make this understand easily, a schematic of workflow should be added in this figure.

For clarity, we agreed to show a schematic of the experimental workflow as a new panel (new Supplementary Fig.9A), see below.

A

DNA concentrations should be written as nM.

Done.

SFig.10: What does the smear on dNTP (+) lane indicate?

This represents replicons, we discussed this point already.

What is the band which appeared above 1000kbp on dNTP (+) lane, and why this appeared?

Indeed, a band between 1 kbp and 1.5 kbp is sometimes observed (see also Fig. 3B and SI Fig. 12B, 13A, 9B). We added the following sentences in the caption of (SI) Fig. 9B to

discuss possible origins of the band: “A lower amplification band was observed between 1 kbp and 1.5 kbp (see also Fig. 3B, Supplementary Fig. 10, 12B, 13A). It is known that the Φ 29 DNA replication system can potentially generate a variety of short DNA products in vitro [16]. Their formation may involve a template-switching mechanism of the DNA polymerase. The produced short DNA molecules can then be replicated to generate short replicons. Alternatively, abortion of DNA replication on a specific sequence or when the polymerase encounters a protein roadblock may cause formation of the observed short DNA product.”

In the legend of Supplementary Fig. 10, we added: “Possible origins of the lower replication band are reported in the legend of Supplementary Fig. 9B.”

DNA concentrations should be written as nM.

Done

SFig. 10: Please explain the working mechanism that the purified p5 increased p3 productivity and the purified p5 and p6 increased p2 productivity.

We presume the reviewer refers to Supporting Fig. 11. We cannot conclude from our data that purified p5 increases p3 productivity and purified p5/p6 increases the yield of p2. The fact that the p2 or/and p3 bands are more intense stems from replication fidelity in the particular conditions.

SFig. 12B: Authors estimated the amount of TP-oriLR-p2-p3 (dNTPs+) from the band intensity of reference DNAs. However, the band intensity of the product is much bigger than references. Thus, the obtained quantification value cannot be reliable.

It is true that in this particular example the sample band is bigger than the reference DNA. In control experiments we had confirmed the linearity of the band ‘volume’ across a wider range of DNA concentrations. Therefore, we can safely extrapolate the curve and give a good estimate of the amount of TP-DNA.

D: Why there are p2 and p3 products in dNTPs (-) control?

It comes from the TP-bound DNA from the first round of replication (not from the 2nd round as no dNTPs was used). Despite its low concentration, this DNA could generate proteins.

E: Is this sample same as that analyzed in B? If so, why smearing and intermediate pattern is different?

No, the two samples are different. The sample in E is a biological repeat (independent preparation of TP-DNA). We indicated in the legend that it was an independent experiment, so we believe there is no ambiguity. In the gels shown in (C) and (F), differences in the DNA patterns for the $-p6-p5$ DNA condition represent regular experimental variability.

SFig. 13: Add the quantification data of the replication in capped and non-capped template DNA.

To be more quantitative, we modified the last sentence in the legend as: “... only the TP-capped substrate is notably amplified both in the absence of *p6-p5* (~10-fold vs. ~2-fold without TP) and in its presence (~4-fold increase from TP-capped DNA).”

Because the concentration of input DNA is now given in nM, it is straightforward for the reader to calculate the amount of synthesized DNA.

SFig. 14: More control data are required, for example, TP-oriLR-p2-p3 DNA (-), p5/p6 (-), etc.

We do not think it is necessary given that we have not observed DNA amplification. We would have certainly considered the suggested controls otherwise.

Reviewer #3 (Remarks to the Author):

In this study titled “Self-replication of DNA by its encoded proteins in liposome-based synthetic cells”, the authors performed DNA self-replication by in vitro-translated phi29 proteins. They successfully translate the proteins, p2, p3, p5, and p6, of the phi29 proteins in the active forms and performed replication of the oriLR-DNA by these proteins. They also showed that the replicated DNA can be a template for the next round of gene expression and replications. Moreover, they performed the gene-expression-coupled DNA replication in liposomes. To date, several groups have tried to constitute translation-coupled DNA self-replication system in vitro, such as phi29 DNAP-cre system and E.coli genome replication system, while recursive DNA replication has not been realized. This study, demonstrating at least three round successive replication, provides a significant advance for the constitution of in vitro DNA replication system. The manuscript was written precisely. Most of the claims the authors raised are convincing with a large amount of supportive data. Therefore, I believe that this study is suitable for publication if the authors appropriately respond the minor points as described below.

Thank you for this positive summary.

Minor points

1. In Figure 1A, I could not understand the difference between “oriLR-DNA” and “(oriLR-)DNA”. Does the “-“ in the latter mean “minus” , not “hyphen”?

Thank you for pointing this out. The brackets indicate that the presence of the *oriLR* sequence is optional). It is clarified in the revised manuscript as follows (legend Fig. 1A): “Alternatively, the expressing DNA does not contain the Φ 29 origin sequences (*oriLR*- in brackets) ...”.

2. Figure 1C. Why does not the phi29 genome contain any genes?

It does, but they are not transcriptionally active in the PURE system. Only the genes under a T7 promoter are expressed and have been annotated. In the legend of Fig. 1C we indicated that only “*the most relevant regulatory elements [were] depicted: ...*” Therefore, we do not find necessary to comment further.

3. Why did the authors use RNeasy column for purification of DNA instead of QIAquick column for DNA?

We used RNeasy column for practical and pragmatic reasons. Initially we only had in house 50-uL elution QIAquick columns, which is too large volume for our DNA purification. Instead, we decided to use the 15- μ L elution RNeasy column, which gave satisfactory results and we kept using this column for the entire project. In the future, we will also try using 15- μ L elution QIAquick columns.

4. Fig 4E is ununderstandable because the third replication was not performed in Figure 4A-D. Fig. 4E should be in the Supplementary Fig. 12, where the replication continued until third round.

The figure is correct and we believe Fig 4E fits here. Fig. 4D shows the protein gel from encoded proteins synthesized from the third IVTT. The TP-*oriLR-p2-p3* DNA illustrated in Fig. 4A is already the replication product of a first IVTTR. It was further used as input DNA for a second IVTTR round (Fig. 4B). Fig. 4D shows the protein products of the third IVTT. For clarity, we completed the legend of Fig. 4E as follows: “A first IVTTR reaction was performed using *oriLR-p2-p3* as input DNA and producing larger amount of TP-*oriLR-p2-p3* (Supplementary Fig. 12B,E). The purified TP-*oriLR-p2-p3* DNA was subsequently used as template for a second IVTTR (B). Finally, the purified DNA products from IVTTR 2 was used for a third IVTT (D).”

5. In page 16, the authors wrote, “Darwinian evolution requires colocalization of the genotypic and phenotypic components”, which is incorrect statement. Evolution of a good template (e.g., a template DNA that can be replicated faster) can evolve without any compartments. For the evolution of encoded proteins, a linkage between genetic molecule and phenotypic molecule (e.g., by compartmentalization) are required.

We agree with the reviewer and modified the sentence accordingly: “Evolution of DNA-encoded proteins requires a linkage between the genetic and phenotypic components.”.

6. Figure 5. It would be helpful to provide the information about how many DNA were in one liposome on average.

We added this information in the legend of the figure: “... synthesized from its gene (7.4 nM bulk concentration of dsDNA) is [...]. Assuming that the entrapped DNA molecules follow a Poisson distribution, liposomes with a diameter of 4 μ m contain ~140 DNA copies on average.”

We also added in the legend of panel (A): “Scale bar is 20 μ m.”

7. Figure 5. I think that the use of acridine orange, which also bind to RNAs in PURE system, cause a high back ground fluorescence in liposomes, which mask the fluorescence of the replicated DNA. I don't think it is necessary for this paper, but it would be better to use other dyes that are more specific to dsDNA, such as SYBR green I.

It is true that AO can also bind RNAs. Our motivations to use AO are described on page 7 in

the Supplementary text, in a dedicated paragraph “Acridine orange as a DNA dye for amplification assays inside liposomes”.

Though it preferentially binds to dsDNA, SYBR green I also binds to ssDNA and RNA. As can be seen in the image below, SYBR green I is significantly bigger than AO, which can be problematic for translocating across the liposome membrane. Nonetheless, we agree that in future studies this option is worth considering. Interestingly, fluorescent spots arising from DNA synthesis in emulsion droplets were also observed using SYBR green I as a DNA dye [new ref. 46]. This earlier observation strongly suggests that the fluorescent spots inside liposomes are not specific to AO. We cited this article and modified a sentence in the Discussion: “Alternatively, accumulation of inorganic phosphate (despite the presence of a pyrophosphatase in the PURE system) during transcription and replication combined with the presence of magnesium ions may trigger the formation of DNA aggregates [46].”

New ref. 46:

Galinis, R., Stonyte, G., Kiseliovas, V., Zilionis, R., Studer, S., Hilvert, D., Janulaitis, A., & Mazutis, L. DNA nanoparticles for improved protein synthesis in vitro. *Angew. Chem.* **128**, 3172–3175 (2016).

.....SYBR green I

acridine orange

8. In the legend of Figure 5, the author wrote, “Due to the stochastic nature of the encapsulation”. This sounds strange because for this size of liposome, stochastic nature would appear only for the components at less than 1 nM. All PURE system components and probably the template DNA are at much higher concentrations.

We think it is correct to refer to the stochastic nature of encapsulation in this context (i.e. not every liposome produces YFP, see the argumentation in the next paragraph). We agree however that other factors may contribute to the observed heterogeneity. Therefore, we tempered the statement and modified the sentence as: “About 30% of the liposomes produce YFP at a detectable level. This functional heterogeneity is probably a consequence of the compositional diversity of the biochemical network within vesicles.”

Bulk DNA concentration was typically 5 nM corresponding to 100 molecules encapsulated on average in liposomes with a diameter of 4 μm (typical size) if one assumes a Poisson distribution. Other components in the PURE system are present at higher concentrations; the protein with lowest concentration is IF1 (100 nM). Even at these concentrations, we argue that stochastic effects due to low-copy number of some entrapped *active* compounds may be significant. First, we do not know the active fraction of DNA, enzymes, protein cofactors and

tRNAs in the stock solutions and to a larger extent inside vesicles. For instance, we showed in a previous study that not all DNA molecules (about 10%) confined in liposomes are transcriptionally active [Nourian and Danelon, 2013. ACS Synthetic Biology 2(4)186-193]. No information is provided by the PURE $_{flex}$ supplier regarding the active fraction of enzymes and cofactors. Second, the above calculation of entrapped DNA molecules is based on the assumption that partitioning of bulk components during liposome formation follows a Poisson behaviour. However, it is probable that the complex – yet unclear – molecular mechanisms of lipid film swelling, accompanying encapsulation and freeze-thaw cycles lead to a distribution of entrapped molecules that deviate from the Poisson prediction. For example, a power-law distribution of ribosomes and other proteins was observed in small vesicles by Luisi and coworkers. Third, about 100 different compounds need to be co-encapsulated above a critical concentration for gene expression to take place. Even in a regime where the individual components are present at >100 copy numbers, stochastic effects will inevitably arise at the level of the reaction network. This is particularly true in biochemical reactions like transcription-translation that exhibit high nonlinearity between the initial concentration of reactants/enzymes and the amount of products.

9. In the legend of Figure 5, the author wrote, “acridine orange can stain the liposome membrane”. Other possibilities are that the RNAs in PURE system (tRNA and mRNA) were stained by acridine orange and attached to the membrane or the membrane fluorescence by Texas-red might leaked. The authors should check these possibilities or just mentioned other possibilities.

We checked possible cross-talk between the Texas-red and the AO channels and could definitely rule out the possibility that Texas-red signal would leak into the AO channel. We agree that we cannot exclude the possibility that AO-bound RNA may interact with the membrane. However, AO bound to RNA should exhibit a larger fluorescence red-shift (Exc./Em. 460 nm/650 nm for RNA vs. 500 nm/526 nm for DNA) that would be incompatible with the settings of the selected green channel. We believe that direct interaction of AO with the bilayer is the most likely scenario. To avoid too much speculation in the main text, we suggest to only mention the latter possibility in the legend of Fig. 5 and to expand discussion in the Supplementary Information to account for the former mechanism. Accordingly, the following sentences have been added in the Supplementary text on page 7: “We noticed that AO could also stain the liposome membrane (Fig. 5B). We checked possible cross-talk between the membrane dye Texas-red and AO channels and could definitely rule out the possibility that Texas-red signal would leak into the AO channel. Due to the hydrophobic nature of AO it is likely that it also partitions in the lipid bilayer. Alternatively, RNAs in PURE $_{flex}$ (mRNA, tRNAs) stained with AO could interact with the liposome membrane. However, we expect that AO bound to RNA exhibits a larger fluorescence red-shift (Exc./Em. 460 nm/650 nm for RNA vs. 500 nm/526 nm for DNA) that would be incompatible with the settings of the selected green channel.”

10. Supplemental Information page 7. “ β Q replicase” should be “Q β replicase”.

Corrected

*** end ***

REVIEWERS' COMMENTS:

Reviewer #2 (Remarks to the Author):

The authors have responded well to the reviewer's questions in the revised manuscript. I think now it's acceptable for publication.

Reviewer #3 (Remarks to the Author):

The authors appropriately answered all of my concerns and questions. I believe the revised manuscript is suitable for publication.